# Optimal Speed Plan for the Overtaking of Autonomous Vehicles on Two-Lane Highways

**Said M. Easa *** and **Maksym Diachuk**

Department of Civil Engineering, Ryerson University, Toronto, ON M5B 2K3, Canada;
maksym.diachuk@ryerson.ca
* Correspondence: seasa@ryerson.ca; Tel.: +1-416-979-5000 (ext. 7868)

**Abstract:** In passing maneuvers on two-lane highways, assessing the needed distance and the potential power reserve to ensure the required speed mode of the passing vehicle is a critical task of speed planning. This task must meet several mutually exclusive conditions that lead to successful maneuvers. This paper addresses three main aspects. First, the issues associated with a rational distribution of the speed of the passing vehicle for overtaking a long commercial vehicle on two-lane highways are discussed. The factors that affect the maneuver effectiveness are analyzed, considering the safety and cost. Second, a heuristic algorithm is proposed based on the rationale for choosing the necessary space and time for overtaking. The initial prediction's sensitivity to fluctuations of the current measurements of the position and speed of the overtaking participants is examined. Third, an optimization technique for the passing vehicle speed distribution during the overtaking time using the finite element method is presented. Adaptive model predictive control is applied for tracking the references being generated. The presented model is illustrated using a simulation.

**Keywords:** autonomous vehicles; speed planning; optimization; required passing time; two-lane highways

---

## 1. Introduction

The concept of autonomous vehicles (AVs) has been under development since the 1990s [1,2], when the first field experiment performed on a freeway was conducted in San Diego, California. Autonomous vehicles have already started to appear on roads across the globe. Clearly, as the AV market expands, transportation professionals and researchers must address an array of challenges before AV becomes a reality. Several government and industry entities have begun to deploy demonstrations and field tests of the technology [3–6]. Centers for testing and validation, education, products, and standards for AV have been established and a variety of conferences are being organized to discuss the planning and modeling for AVs [6]. Currently, researchers, scientists, and engineers are investing significant resources to develop supporting technologies. This vibrant state of development has motivated the proposed program.

Autonomous vehicles have numerous advantages [7]. Since more than 90% of fatal vehicle collisions are attributed to human error, AV technologies can substantially reduce deaths and injuries from vehicle collisions. Autonomous vehicles can also reduce traffic congestion, reduce carbon dioxide emissions, increase the highway capacity, reduce fuel consumption, improve public transportation services, and provide more efficient parking. On two-lane highways, the driver's decision to overtake may be risky, since a driver has to judge the operational behavior of opposing and impeding vehicles and decide to pass in a very short amount of time. In addition, the speed plan of the passing vehicle is based on the driver's perception of its dynamic properties.

---

In autonomous overtaking, prediction, decision making, and continuous monitoring of vehicle performance are performed by an onboard system. The operation of autonomous vehicles is based on planning the movement trajectory, as well as other reference trajectories. The system searches for an optimal way to bypass obstacles while maintaining traffic safety. For overtaking on two-lane highways, the vehicle's on-board system must estimate the distance and time required for the passing maneuver and distribute the passing vehicle speed and steering angle based on conflicting criteria. Therefore, the optimization of passing vehicle control for safe and efficient passing maneuvers of autonomous vehicles can be achieved.

Numerous researchers have addressed the issues of planning motion reference lines and control parameters for autonomous vehicles and attempted to find the best trajectories, speed plans, and state-space sequences. Schwarting et al. [8] considered the concept of parallel autonomy, where autonomous control works as an option to monitor and correct driver errors (called shared control). The work initially focused on curvilinear road profiles for which the optimal trajectory of motion was determined using nonlinear model predictive control (MPC), which allowed a consideration of turns and avoidance of moving and static obstacles. Both kinematic and dynamic models were used as vehicle models. The optimization model included a probabilistic collision estimate, and geometric and physical (tire-road adhesion) constraints. The intervention parameter was used to assess the degree to which the system is involved in driver actions. The optimization algorithm provided fast convergence. The only limitation of this work is that the acceleration input parameter is not related to the engine's potential.

Talamino and Sanfeliu [9] presented a technique for planning a movement trajectory and speed plan of an autonomous vehicle in urban areas based on $G^2$-splines. The polynomial fitting involved iterations equivalent to the optimization of curvature parameters. To simulate a sufficiently long path, a fifth-degree polynomial was used. However, such polynomials are often unstable between nodes. For the speed distribution, a third-degree polynomial was proposed, where the transition time was determined based on the values of parameters (speeds and accelerations) at the end points. Acceleration was limited to the maximum value and was not related to the parameters of the power plant. Not enough information about the overtaking maneuver parameters was provided. González et al. [10] reviewed the methods used to plan autonomous vehicle movement. Graph algorithms were mainly used to determine the minimum maneuver path on the surrounding space grid. The State Lattice algorithm executed path searching using state-space mesh generation. Sampling-Based Planners generated random state-spaces and looked for their ties. A rapidly exploring random tree (RRT) made it possible to use structured spaces. Lines and circles, closed curves, polynomial curves, Bézier curves, and spline curves were used to represent the path forecast. A numerical method for optimizing a function subject to different constraints was used.

Gu et al. [11] proposed a planning method that automatically discovers tactical maneuver patterns and fuses pattern reasoning trajectories based on the idea of using pseudo-homology along with characterizing workspace regions. Different patterns can be extracted, depending on the spatial area where the trajectory terminates (region-based distinction), how it gets there around the obstacles (homology-based distinction), and what overtaking (if any) order it follows (sequence-based distinction). A series of virtual tests were conducted and confirmed the effectiveness of the method. Wang et al. [12] considered the process of building an optimal overtaking route based on minimizing the probability of the vehicles' presence in the area with close coordinates. An integration process was used to solve the nonlinear optimization. Kala and Warwick [13] considered the process of overtaking based on the conditions of maximum speed movement, but limited acceleration. A speed plan was not considered for overtaking. Testing of the model was carried out at low speeds, with a large distance between the approaching vehicles.

Babu et al. [14] presented an MPC framework based on path speed decomposition for autonomous driving. The concept of a time scaled collision cone, which constrains and formulates forward-speed quadratic optimization, was presented. Collision modeling between rectangular objects

was presented. The planned vehicle was reduced to a point and the dynamic obstacle was enlarged using the concept of the Minkowski sum. The autonomous driving scenarios were validated with computations of lane change, overtaking, and merging maneuvers among multiple dynamic obstacles. Tomas-Gabarron et al. [15] considered how to trace the optimum trajectory of a high-speed vehicle that changed its lateral position within a time interval. Four different functions were proposed, along with their relative merits. The presence of Gaussian noise in the sensors' measurements was studied regarding its influence on the final trajectories. Different performance criteria for the optimization of such maneuvers were presented, as well as an analysis on how path deviations can be minimized by using trajectory smoothing techniques, such as the Kalman filter. Liu et al. [16] focused on speed profile planning for a given path represented by a set of waypoints. The speed profile was generated using temporal optimization that searched the time stamps for all waypoints. Non-convex temporal optimization was approximated by a set of quadratic programs that were solved iteratively using a slack convex feasible algorithm to speed up computations. Other interesting developments of motion planning and control can be found elsewhere [17–21].

This paper presents a new technique of speed planning for the overtaking of autonomous vehicles on two-lane highways. The methodology consists of two main analytical tools. The first tool is a heuristic algorithm that determines the time and distance required for a safe passing maneuver. The algorithm relies on uncertainty-based thresholds of the opposing and impeding vehicles and the minimum and maximum performances of the passing vehicle. The second tool is a quadratic optimization model that determines the optimal speed distribution to ensure a smooth path for the passing vehicle. If needed, the speed plan is updated during the maneuver. The proposed method focuses on finding a rational scheme for vehicle limiting performances. The method aims to simultaneously satisfy several competing objectives, such as a sufficient overtaking time, trajectory smoothness, energy consumption, and collision avoidance. The reason for implementing this approach is that if overtaking is performed with the maximum vehicle performance, there will be a risk of losing lateral stability (due to random external forces) and energy consumption will be high. However, this scenario provides the minimum time and reduces the probability of a head-on collision. On the other hand, if overtaking is performed slowly, a good stability and controllability will be ensured, but there may not be enough of a safety margin at the end of the maneuver.

## 2. System Description

### 2.1. Overtaking Phases

The phases of overtaking on two-lane highways are shown in Figure 1. In Phase *a* (*obstacle rear reach*), the passing vehicle will start to drive into the oncoming traffic lane (State 1) and reaches the rear edge of the impeding vehicle (State 2), as shown in Figure 1a. Therefore, the longitudinal component of the passing vehicle path during the *bypassing* phase is $X_{pb}$, and during this time, the impeding vehicle travels the distance $X_{ib}$. In Phase *b* (*obstacle front reach*), the passing vehicle travels from the *bypass point* (State 2) to the *critical point* (State 3), where its front aligns with that of the impeding vehicle. During this time, the passing and impeding vehicles travel distances $X_{po}$ and $X_{io}$, respectively.

In Phase *c* (*maneuver completion*), the passing vehicle passes the longitudinal distance $X_{pf}$ to ensure an adequate safe distance $d_f$ between itself and the impeding vehicle, traveling the distance $X_{if}$, as shown in Figure 1c. The passing and opposing vehicles must provide a safety margin distance $X_m$ between their fronts which is equivalent to the minimum safety margin time $t_{mm}$. Note that in this phase, the lateral movement back to the original lane will not start until the passing vehicle has already overtaken the impeding vehicle.

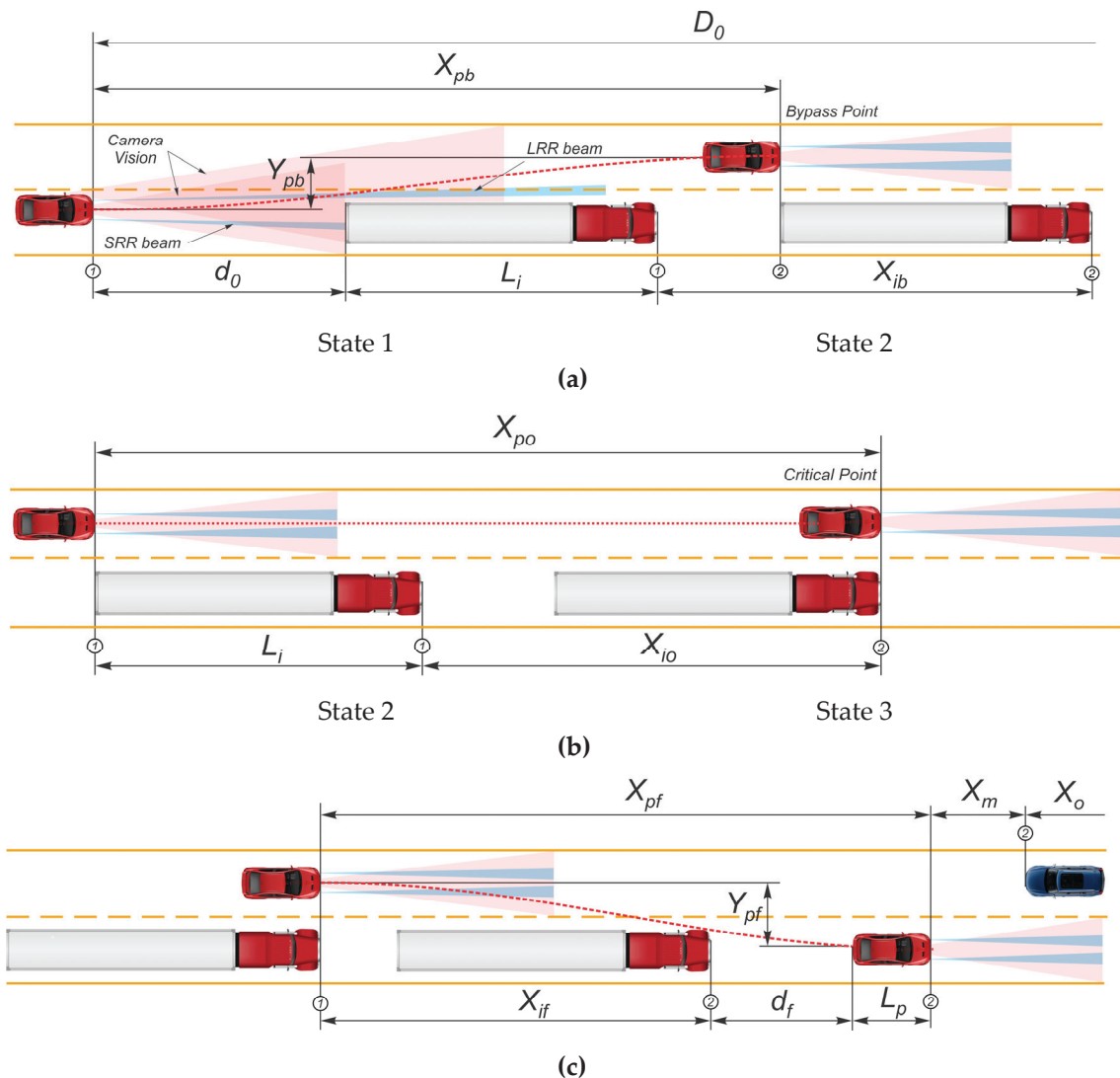

**Figure 1.** Phases of overtaking: (**a**) obstacle-rear reach, (**b**) obstacle-front reach, and (**c**) maneuver completion.

## 2.2. Assumptions

The developed system involves the following assumptions:

(a)   The roadway is assumed to be straight, with ideal surface friction and no external forces, such as gusts of wind. In addition, the road is assumed to have very gentle vertical curvature that would not obstruct sensor measurements;

(b)   Before the overtaking maneuver starts, it is assumed that the estimation of the vehicles' positions has already been carried out, the forecast has been made, and the passing vehicle is ready to start the passing maneuver. In this state, the distance $D_0$ between the passing and opposing vehicles and the distance $d_0$ between the impeding and passing vehicles are estimated using long- and short-range radars, respectively;

(c)   The length of the impeding vehicle is estimated using machine vision technology. The sensor measurements (distance and angle) are assumed to be independent;

(d)   The AV is assumed to possess the input data required for the proposed model: roadway data (e.g., lane width and speed limit) and vehicle characteristics (e.g., acceleration–speed relation);

(e)   Only the passing vehicle is autonomous, and the impeding and opposing vehicles are assumed to be human-driven vehicles. The overtaking maneuver involves only one impeding vehicle;

(f)　In the heuristic algorithm, it is assumed that the actual speed fluctuations detected by the radars remain within the uncertainty thresholds, for which the forecast is considered reliable. Therefore, the changes of vehicle locations will occur within the boundaries determined by the specified measurement thresholds. If the threshold is exceeded, the forecast should be recalculated;

(g)　Uncertainty is considered in the estimation of the speeds of the passing and impeding vehicles. Based on the uncertainty thresholds, possible violations of the thresholds are analyzed. No uncertainty propagation throughout the prediction process is considered in this study;

(h)　The algorithm continuously works with the vehicle sensory system within the maneuver and only estimates new overtaking parameters when thresholds are violated, thus helping to reduce the load on the on-board computer system.

### 2.3. Logic of the Speed Control Model

The logic of the speed control model is shown in Figure 2. The model involves four main tasks. First, the operational thresholds of the passing, opposing, and impeding vehicles are established. These thresholds include (1) uncertainty-based thresholds for the predicted speeds of the opposing and impeding vehicles, (2) the minimum performance limit of the passing vehicle, and (3) the maximum performance limit of the passing vehicle. Second, the speed profiles of the three involved vehicles are established and initial values of the time $t_p$ and distance $X_p$ required for safe maneuver completion are established using a heuristic algorithm, as described later. These variables are used for determining the optimal distribution of speeds and trajectory planning of the maneuver. Third, quadratic optimization is used to develop a smooth curve for the path of the passing vehicle that can serve as a reference for the control law implementation during maneuver realization.

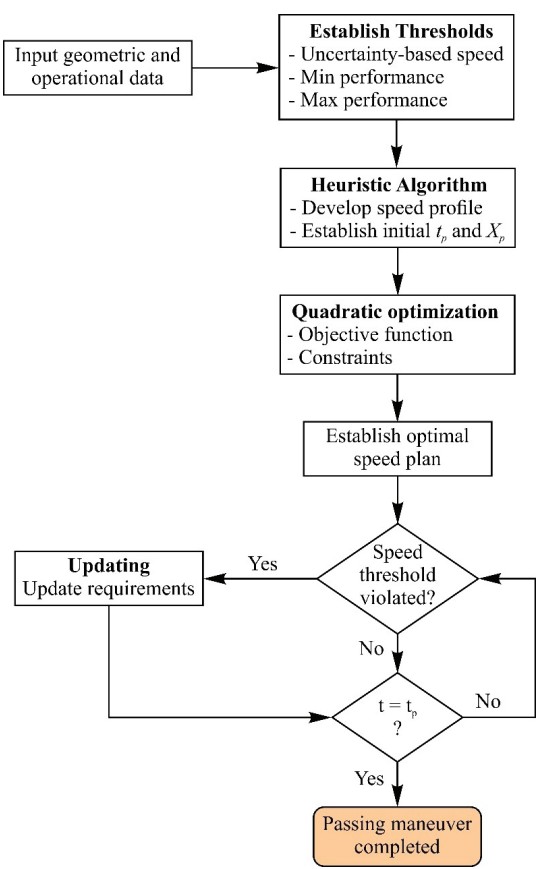

**Figure 2.** Logic of the proposed speed control model for the overtaking of autonomous vehicles.

During maneuver execution, if the upper (in terms of speed values) confidence thresholds of the predicted speed of the opposing or impeding vehicles are violated, the time and distance required for completing the maneuver safely are updated. As noted in Figure 2, if $t < t_p$, the system continues with the last prediction and is updated if the speed thresholds are violated. Otherwise, the maneuver is successfully completed.

## 2.4. Establishing Operational Speed Thresholds

The measured speeds of the opposing and impeding vehicles have uncertainty. The speed of the opposing vehicle is estimated using radar sensors located in the passing vehicle. Using four signals, $\Delta t$ apart, four distances to the opposing vehicle $d_i$ and the corresponding azimuth angles $\theta_i$ are recorded, where the polar coordinates used with the origin point lie with the sensor location. The deterministic distance crossed by the vehicle in consecutive time intervals was presented by Hassein et al. [22] (2018):

$$dV_{opp_i} = \sqrt{d_i^2 + d_{i+1}^2 - 2 \cdot d_i \cdot d_{i+1} \cdot cos(\theta_{i+1} - \theta_i)} - dp_i, \ i = 1 \ to \ 3, \tag{1}$$

where $dV_{oppi}$ denotes the distance traveled by the opposing vehicle during $\Delta t$ and $dp_i$ denotes the distance traveled by the passing vehicle during $\Delta t$. The speed of the opposing vehicle, $V_{opp}$, can then be derived as

$$V_{opp} = \frac{1}{3 \cdot \Delta t} \cdot dV_{opp_1} - \frac{7}{6 \cdot \Delta t} \cdot dV_{opp_2} + \frac{11}{6 \cdot \Delta t} \cdot dV_{opp_3}. \tag{2}$$

Let the errors in $d_i$ and $\theta_i$ measurements of the radar be denoted by $e_d$ and $e_a$, respectively. Then, these errors will propagate and produce an error in $V_{opp}$ of Equation (2). To calculate this error, let the four variables of Equation (1) ($d_i$, $\theta_i$, $d_{i+1}$, and $\theta_{i+1}$) be denoted by $x_i$, where $i$ denotes 1 to 3. Using the Taylor series, the standard deviation of $Y$, $\sigma_y$, was given by Benjamin and Cornell [23]:

$$\sigma_y = \sqrt{\sum_{i=1}^{n} \left(\frac{\partial f}{\partial x_i}\right)^2 \cdot \sigma_{xi}^2}, \tag{3}$$

where $\sigma_{xi}$ denotes the standard deviation (SD) of random variable $x_i$. By applying Equation (3) to Equation (1), the standard deviation of $dV_{oppi}$, $\sigma_{dVoppi}$, can be derived as

$$\sigma_{dV_{opp_i}} = \sqrt{\frac{k_d \cdot e_d^2 + k_a \cdot e_a^2}{dV_{opp_i} + dp_i}}, \tag{4}$$

where

$$k_d = \left(\left(d_i^2 + d_{i+1}^2\right) \cdot \left(1 + cos^2(\theta_{i+1} - \theta_i)\right) - 4 \cdot d_i \cdot d_{i+1} \cdot cos(\theta_{i+1} - \theta_i)\right),$$
$$k_a = 2 \cdot d_i^2 \cdot d_{i+1}^2 \cdot sin^2(\theta_{i+1} - \theta_i).$$

Then, by applying Equation (3) to Equation (2), the standard deviation of the speed of the opposing vehicle, $\sigma_{Vopp}$, is obtained as

$$\sigma_{V_{opp}} = \sqrt{\left(\frac{1}{3 \cdot \Delta t}\right)^2 \cdot \sigma_{dv_1 - opp_1} + \left(\frac{7}{6 \cdot \Delta t}\right)^2 \cdot \sigma_{dv_2 - opp_2} + \left(\frac{11}{6 \cdot \Delta t}\right)^2 \cdot \sigma_{dv_3 - opp_3}}. \tag{5}$$

Similarly, the deterministic distance travelled by the impeding vehicle between consecutive time intervals before overtaking is

$$dV_{imp_i} = d_{i+1} - d_i + dp_i, \ i = 1 \ to \ 3. \tag{6}$$

The standard deviation of the distance $dv_i - imp_i$, $\sigma_{dVimpi}$, of Equation (6) is given by

$$\sigma_{dV_{imp_i}} = \sqrt{d_i^2 + d_{i+1}^2}. \tag{7}$$

Then, the speed of the impeding vehicle $V_{imp}$ and its standard deviation $\sigma_{Vimp}$ are calculated using Equations (2) and (5), after replacing $dV_{oppi}$ with $dV_{impi}$ ($i$ denotes 1, 2, and 3).

For the 95% confidence level, the true speed of the opposing or impeding vehicle lies within approximately two standard deviations from the measured value. Therefore, the confidence ranges used in the proposed heuristic algorithm for the opposing and impeding speeds are calculated as $V_{opp} \pm 2 \cdot \sigma_{Vopp}$ and $V_{imp} \pm 2 \cdot \sigma_{Vimp}$, respectively.

## 3. Heuristic Algorithm

### 3.1. General

The heuristic algorithm estimates the rational time and distance required for overtaking. The logic of the algorithm is presented in Figure 3. For each relative position of the vehicles involved in the overtaking maneuver, a combination of measurements will be unique, and the number of possible maneuvers will be numerous. Suppose that at a certain time, the speed values (sample) of the impeding and opposing vehicles have been estimated using the passing vehicle sensory system (noise variances are supposed to be known). At the initial time, the impeding and opposing vehicles are located at distances $d_{(-m)}$ and $D_{(-m)}$, respectively, relative to the passing vehicle, where the index $(-m)$ means the backward number of radar measurement cycles required before the prediction is made. Considering the time required for prediction, the values at the maneuver's beginning time $t_0$ will become $d_0$ and $D_0$, respectively. If the opposing and impeding vehicles continue their motions at speeds close to the measured ones, then the change in their positions will be approximately linear, which determines the slope ($dX/dt$) of the corresponding curves (blue and green curves in Figure 3). In turn, the speed measurements also have uncertainty.

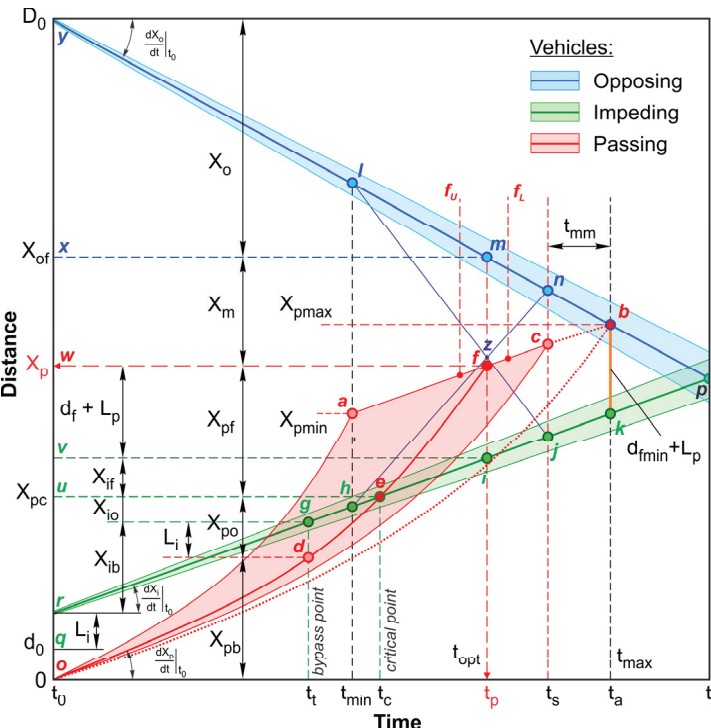

**Figure 3.** Logic for selecting rational values of time and distance for overtaking.

Point *a* ($t_{min}$, $X_{pmin}$) corresponds to the minimum time and distance, while Point *b* ($t_{max}$, $X_{pmax}$) corresponds to the maximum time and distance. Consider the vehicle's maximum performance with a full fuel supply along the *oa* curve. Considering the mean performance at Point *f* ($t_p$, $X_p$), a distance ($d_f + L_p$) would be needed to complete the maneuver, where the linear segment *ab* represents a set of solutions that correspond to the desired time and distance of overtaking. To predict the minimum distance required for overtaking completion, the passing vehicle speed at the maneuver's end, and the minimum safe distance between the passing and impeding vehicles (Figure 1c), the space $d_{fmin}$ (Figure 3) should be predetermined.

The safety margin $t_{mm}$ guarantees a distance between the passing and opposing vehicles after maneuver completion. This corresponds to Point *c* on the segment *ab*. The *oc* curve will represent the lower boundary of the field *oac* of the valid time–distance realizations. If the speed changes of the opposing and impeding vehicles remain within the threshold values, the lower boundary of the opposing vehicle's distance dependency with the basis (instant mean) line *yp* will not reach Point *c*, maintaining the safety margin until time $t_s$. This may be the key point for determining the threshold conditions.

The main idea of the rational point search is to simultaneously meet the criteria of the safety margin and vehicle performance reserve for unpredicted circumstances. Suppose the opposing vehicle moves according to the nominal straight line *yp*, and the impeding vehicle moves along the nominal straight line *rp*. Then, the nominal distance ($d_f + L_p$) for completing the maneuver (segment *bk*) will correspond to $t_a$. Considering $t_{mm}$, the new boundary will imply the safety limit at time $t_s$. Now, it is necessary to choose a point on segment *ac* that would meet the required criteria. There are many approaches that can be employed to achieve this. One of the possible approaches is to use the trapezoid *hlnj* to proportionally split segment *ac*. That is, the vertical lines of the intersection Point *z* of the trapezoid *hlnj* diagonals correspond to intersect line *ac* at Point *f*. Segments *fm* and *fi* characterize the distances to the opposing and impeding vehicles, respectively. In this case, Point *f* determines the $t_p$ and $X_p$ required for the passing vehicle. At this moment, the distance traveled by the opposing vehicle is $X_o$, which corresponds to its final position $X_{of}$ (Point *x*). This approach ensures a stable and gradual redistribution of Point *f* by increasing the minimum safety margin.

The feasible vehicle acceleration performance corresponds to an upper limit *oa* and a lower limit that achieves the safety margin *oc*. The search for rational values and the law of speed change in the third phase (Figure 1c) depends on the difference between the speeds of the passing and impeding vehicles, which may exhibit values from the admissible minimum to the maximum being stipulated by the full performance mode (i.e., the upper limit *oa* at the beginning and completion of the lane change, Figure 3). Obviously, there is a need to optimize the movement trajectory in such a way as to ensure that both criteria (safety and power margins) account for possible changes in maneuver conditions (e.g., vehicle speeds and/or unpredictable forces).

Note that the geometric average scheme of Figure 3 is justifiable from two perspectives. First, when overtaking with full use of the engine energy, there is no more power reserve for unforeseen circumstances (e.g., gusts of wind, random forces, or control system failure). That is, in a critical situation, it would not be possible to compensate for their influence, especially if one (or both) of the other two vehicles suddenly increases their speed. Second, the maximum performance modes are undesirable in terms of the vehicle lateral stability and lateral sliding. If the speed is high, a stable lane change requires a greater distance, according to the criterion of limited lateral acceleration.

### 3.2. Vehicle Performance Thresholds

The lower limit of the vehicle acceleration performance is determined based on the velocity plan ensuring the minimum safety. The criterion of the minimization of energy consumption can be applied to the search for the time–distance curve adjusted for the curvilinear trajectory of the vehicle. The upper limit (curve *oa*) represents the vehicle potential provided with a full fuel supply under the ideal conditions of motion. The definition of the upper limit can be based on the characteristic of the

vehicle dynamic factor (specific free traction force) restricted by the conditions of road surface adhesion and the reduced total movement resistance, including the road macro-profile. This upper limit may be estimated using mapping and GPS. The excess of the dynamic factor can be employed to accelerate the vehicle. Therefore, it is possible to build the speed–time and time–distance dependences for the acceleration mode, by which the necessary overtaking time and distance can be determined using the iterative method considered by Diachuk at al. [24]. The obtained values will represent the vehicle performance on a straight road section. To account for the curvilinear trajectory of the maneuver, the values of the time–distance curve can be adjusted.

An important stage of the forecasting is determination of the maximum vehicle acceleration capabilities (speed-acceleration). For this purpose, it is necessary to have a diagram of the free traction force (air resistance is subtracted) or a diagram of the dynamic factor (specific free traction force). If there is access to GPS signals and digitized terrain maps, it is possible to track the motion condition changes to correct control signals (throttle). In addition, a diagram of possible vehicle accelerations in the current conditions (Figure 4) is necessary considering movement resistance (surface quality) and adhesion to the road surface (weather conditions), since restrictions are needed for the optimal distribution of the speed plan. If the initial speed and the average speed are known, then for the linear constraints used in linear quadratic programming (LQP), the limiting values of acceleration achievable under given conditions can be determined.

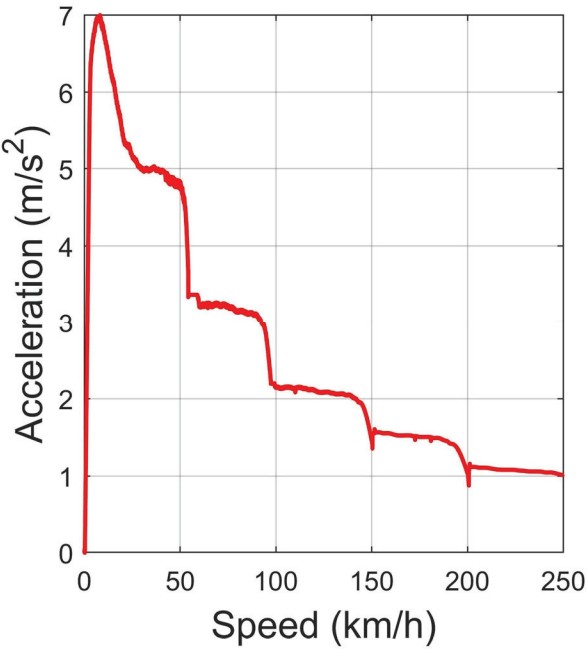

**Figure 4.** Vehicle acceleration as a function of speed (example of Audi A4 Quattro).

If the measurements are known and the time–distance curve is determined for the case of the passing vehicle's maximum performance [24], it is possible to immediately define whether the maneuver is feasible. For this, it is necessary to set, rightward of Point $a$ (Figure 5), the safety margin $t_{mm}$ to Point $b$. The ratio of the inclination angles of opposing and impeding vehicles' curves, providing an intersection in Point $p$ and adequate space $d_f + L_p$ to complete the maneuver, will ensure the time limit and the possible path of the maneuver. It is obvious that with a larger inclination angle of linear prediction for the opposing vehicle, the sensitivity's influence on the remaining time safety margin $t'_{mm}$ is decreased. This does not mean that the vehicle will not be able to use the power corresponding to the segment $ab$; however, the maneuver execution in this mode will be associated with a decrease in the guaranteed level of safety. Therefore, the condition of the maneuver possibility is $t_a > t_{min} + t'_{mm}$, where $t'_{mm} \leq t_{mm}$ due to the lower sensitivity.

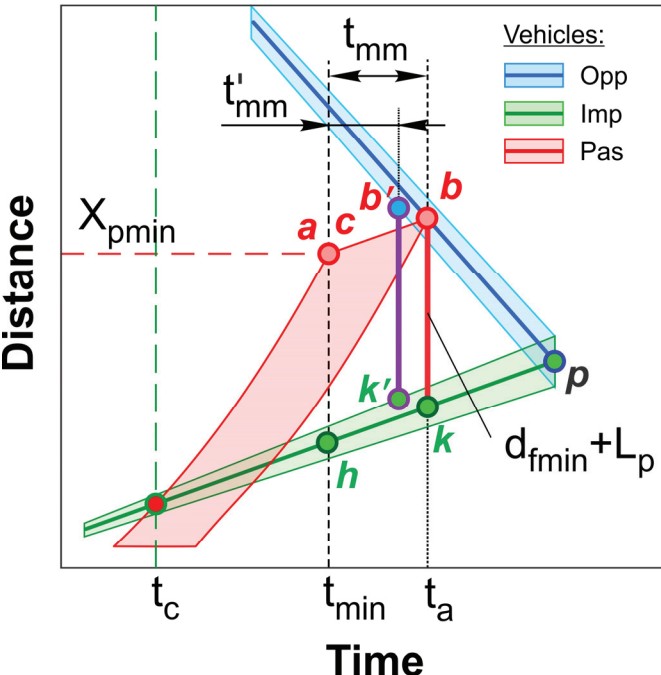

**Figure 5.** Limiting case of the maximum performance.

Another important point concerns the maximum allowable time required for overtaking. It is also possible that the time range $t_{min} - t_l$ according to the measured and evaluated data will be quite wide, which may lead to the calculation of large values of $t_p$ having no real meaning because of predicting a protracted maneuver. In this regard, the maximum maneuver time should also be limited based on the ratio of the times $t_{min}$ and $t_a$, $t_l$.

### 3.3. Mathematical Formulation

To describe the movement laws of overtaking using the performance curve *of*, suppose that the measurements are evaluated at intervals of $\Delta t_m$ based on the preset frequency of the radar system. Then, the current discrete time will be $n \cdot \Delta t_m$, where $n$ takes both positive and negative values relative to the starting point of the maneuver (sample $n = 0$). The results of the relative measurements at time $t_n$ are as follows:

$$\left( \begin{array}{cccccc} D|_{t_n} & v_o|_{t_n} & d|_{t_n} & v_i|_{t_n} & X_p\big|_{t_n} & V_p\big|_{t_n} \end{array} \right) = \left( \begin{array}{cccccc} D_n & v_{on} & d_n & v_{in} & X_{pn} & V_{pn} \end{array} \right), \tag{8}$$

where $D_n$ denotes the current measured distance between the passing and opposing vehicles; $v_{on}$ denotes the current measured opposing vehicle speed; $d_n$ denotes the current measured distance between the passing vehicle and the rear of the impeding vehicle; $v_{in}$ denotes the current measured impeding vehicle relative speed; and $X_{pn}$ and $V_{pn}$ denote the estimated current self-position and speed of the passing vehicle, respectively.

Nevertheless, the speed forecast processing requires more time $\Delta t_{pr}$ and computational resources. In this regard, the algorithm should be organized to avoid frequent recalculations that do not significantly affect the quality of the forecast. Therefore, the time $\Delta t_{pr}$ must be a multiple of the time $\Delta t_m$ ($\Delta t_{pr} = m \cdot \Delta t_m$), where $m$ is the factor of cycle multiplicity. This is provided that to process the forecast, the time $\Delta t_{pr} = t_0 - T_{-1}$ is needed (where $T_{-1}$ is the time before the forecast is made until $t_0$) and that during this period, the vehicle speed does not change significantly, i.e.,

$$\left. \frac{dX_p}{dt} \right|_{T_{-1}} \approx \left. \frac{dX_p}{dt} \right|_{t_0}, \left. \frac{dX_o}{dt} \right|_{T_{-1}} \approx \left. \frac{dX_o}{dt} \right|_{t_0}, \left. \frac{dX_i}{dt} \right|_{T_{-1}} \approx \left. \frac{dX_i}{dt} \right|_{t_0} \tag{9}$$

For the opposing and impeding vehicles, linear predictions can be made from a tangent angle $k$ and the values at the point as $(y - y_0) = k (x - x_0)$. The potential global positions of the opposing and impeding vehicles are determined relatively through the predicted movement of the passing vehicle. Therefore, while performing the maneuver, the current state vector assessment $(X_{pn}, V_{pn})^T$ is periodically recalculated based on the sensor fusion technology. Then,

$$\begin{pmatrix} X_{on} \\ X_{in} \end{pmatrix} = \left( \begin{pmatrix} D_n & v_{on} \\ d_n + L_i & v_{in} \end{pmatrix} + \begin{pmatrix} X_{pn} & V_{pn} \\ X_{pn} & V_{pn} \end{pmatrix} \right) \cdot \begin{pmatrix} 1 \\ t - t_n \end{pmatrix} \tag{10}$$

To determine the locking time $t_l$ within the overtaking pocket, the condition for the intersection of linear predictions at Point $p$ corresponding to $t_l$ is $X_o = X_i$. Note that $V_o$ and the time at the moment before the forecast are negative, thus $T_{-1} = -\Delta t_{pr} = -m \cdot \Delta t_m$. Consequently, the initial measurement is carried out in $m$ cycles before the maneuver starts. That is,

$$\begin{pmatrix} 1 \\ -1 \end{pmatrix}^T \cdot \begin{pmatrix} X_o \\ X_i \end{pmatrix} = \begin{pmatrix} 1 \\ -1 \end{pmatrix}^T \cdot \left( \begin{pmatrix} D_{(-m)} & v_{o(-m)} \\ d_{(-m)} + L_i & v_{i(-m)} \end{pmatrix} + \begin{pmatrix} X_{p(-m)} & V_{p(-m)} \\ X_{p(-m)} & V_{p(-m)} \end{pmatrix} \right) \cdot \begin{pmatrix} 1 \\ t_l + \Delta t_{pr} \end{pmatrix} \tag{11}$$

The passing vehicle position at time $T_{-1}$ relative to $t_0$ is estimated as $X_{p(-m)} \approx -V_{p(-m)} \cdot \Delta t_{pr}$.

$$t_l = \frac{D_{(-m)} - d_{(-m)} - L_i}{v_{i(-m)} - v_{o(-m)}} - m \cdot \Delta t_m \tag{12}$$

To determine $t_a$, the minimum distance $X_o - X_i = d_{fmin} + L_p$ is the distance between the impeding and passing vehicles. Therefore, $t_a$ can be defined by the difference $(d_{fmin} + L_p)$ between the functions $yp$ and $rp$, similar to Equation (12). That is,

$$t_a = t_l - \frac{d_{fmin} + L_p}{v_{i(-m)} - v_{o(-m)}} \tag{13}$$

To determine Point $z$ of the diagonals' intersection, the distances at Points $l$, $h$, $n$, and $j$ corresponding to $t_{min}$, $t_s$, using Equation (12), are

$$\begin{pmatrix} X_l & X_n \\ X_h & X_j \end{pmatrix} = \begin{pmatrix} D_{(-m)} - V_{p(-m)} \cdot \Delta t_{pr} & V_{p(-m)} + v_{o(-m)} \\ d_{(-m)} + L_i - V_{p(-m)} \cdot \Delta t_{pr} & V_{p(-m)} + v_{i(-m)} \end{pmatrix} \begin{pmatrix} 1 & 1 \\ t_{min} + \Delta t_{pr} & t_s + \Delta t_{pr} \end{pmatrix} \tag{14}$$

The values of the coefficients $k$ and $b$ are determined from the matrix relations:

$$\begin{pmatrix} X_l & X_h \\ X_j & X_n \end{pmatrix} = \begin{pmatrix} t_{min} & 1 \\ t_s & 1 \end{pmatrix} \begin{pmatrix} k_{lj} & k_{hn} \\ b_{lj} & b_{hn} \end{pmatrix} \text{ and } \begin{pmatrix} k_{lj} & k_{hn} \\ b_{lj} & b_{hn} \end{pmatrix} = \frac{1}{t_s - t_{min}} \cdot \begin{pmatrix} -1 & 1 \\ t_s & -t_{min} \end{pmatrix} \begin{pmatrix} X_l & X_h \\ X_j & X_n \end{pmatrix} \tag{15}$$

The intersection condition at $t_p$ is the equality of the ordinates of segments $X_{lj}$ and $X_{hn}$:

$$X_{lj} - X_{hn} = \begin{pmatrix} t_p \\ 1 \end{pmatrix}^T \cdot \begin{pmatrix} k_{lj} \\ b_{lj} \end{pmatrix} - \begin{pmatrix} t_p \\ 1 \end{pmatrix}^T \cdot \begin{pmatrix} k_{hn} \\ b_{hn} \end{pmatrix} = \begin{pmatrix} t_p \\ 1 \end{pmatrix}^T \cdot \begin{pmatrix} k_{lj} - k_{hn} \\ b_{lj} - b_{hn} \end{pmatrix} = 0 \text{ and } t_p = \frac{b_{lj} - b_{hn}}{k_{hn} - k_{lj}} \tag{16}$$

The equation of a line passing through the segment $ab$ can be determined based on the coordinates of two points:

$$\begin{pmatrix} X_a \\ X_b \end{pmatrix} = \begin{pmatrix} t_{min} & 1 \\ t_a & 1 \end{pmatrix} \cdot \begin{pmatrix} k_{ab} \\ b_{ab} \end{pmatrix} \text{ and } \begin{pmatrix} k_{ab} \\ b_{ab} \end{pmatrix} = \begin{pmatrix} t_{min} & 1 \\ t_a & 1 \end{pmatrix}^{-1} \cdot \begin{pmatrix} X_a \\ X_b \end{pmatrix} \tag{17}$$

Therefore, the distance $X_p$ corresponding to time $t_p$ is

$$X_p = \begin{pmatrix} t_p \\ 1 \end{pmatrix}^T \cdot \begin{pmatrix} k_{ab} \\ b_{ab} \end{pmatrix} = \frac{1}{t_a - t_{min}} \cdot \begin{pmatrix} t_p \\ 1 \end{pmatrix}^T \cdot \begin{pmatrix} -1 & 1 \\ t_a & -t_{min} \end{pmatrix} \begin{pmatrix} X_a \\ X_b \end{pmatrix} \tag{18}$$

The aim of this stage is to obtain the values of $t_p$ and $X_p$ (Figure 3), which can be used for the optimal distribution of speeds and trajectory planning of the maneuver.

## 4. Quadratic Optimization Model

### 4.1. General

After the rational values of $t_p$ and $X_p$ are determined using the heuristic algorithm previously described, the desired trajectory of motion is determined using a kinematic model. Such a model makes path planning simpler and faster and provides a smooth curve that can be adjusted, depending on the priorities of the kinematic parameters. This curve can serve as a reference for the control implementation of laws of the autonomous vehicle during maneuver realization.

Suppose that the overtaking maneuver is being planned for a relatively straight road section. Then, the formation of the longitudinal and transversal components of the speed plan can be considered independent. Consider the process of forming the longitudinal component of vehicle speed according to the direction of road marking lines (Figure 6). The values of the time and distance required for overtaking ($t_p$, $X_p$) are determined before the maneuver starts at time $t_0$. Obviously, there are many realizations of the distribution of the speed's longitudinal component, such that their integral over the time interval ($t_p - t_0$) equals the distance $X_p$. These curves will at least differ in the value of $V_X$ and its derivative $dV_X/dt$ (acceleration) at the nodal points.

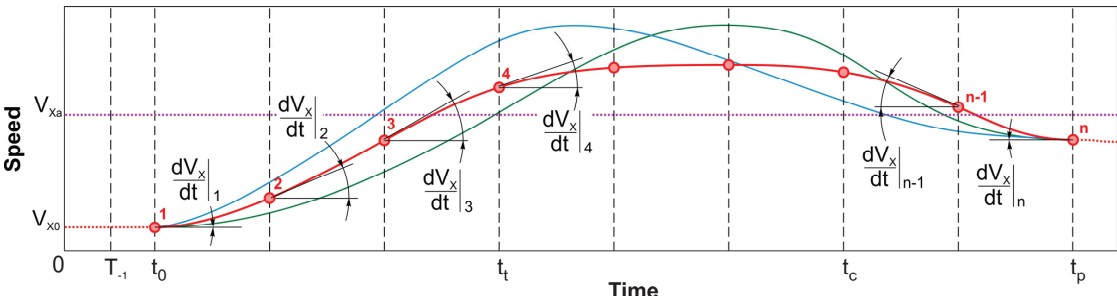

**Figure 6.** Different speed plan distributions with fixed values of $t_p$ and $X_p$.

The speed distribution must also satisfy various requirements, such as technical, operational, economic, and safety requirements, which may be conflicting. The technical requirements are associated with the propulsion system to ensure the required vehicle's performance under external constraints, such as a slope, road resistance, tire adhesion, and head wind. That is, the curvature of the speed plan should be comparable to the curvature of the output characteristics of a power unit operating in the transient mode. The operational requirements ensure that the traction and steering controls are smooth by imposing restrictions related to the vehicle's steerability and stability. Therefore, significant speed and acceleration are undesirable at the moments of lane change. The economic requirements aim to minimize the work of the vehicle power unit by minimizing vehicle acceleration, since more fuel consumption occurs at the moments of speed change. From this perspective, the cumulative derivative of the speed plan curve should also be minimal. The safety requirements limit the maximum speed, acceleration, angular speed of the steering wheel, and time spent near the impeding vehicle.

*4.2. Objective Function*

The objective function of the model minimizes the speed variation, acceleration variation, and sharpness (differences between tangent coefficients of adjacent nodes). Therefore, the optimal speed distribution is written as

$$\min_{V_X} J = W_V \cdot J_V + W_A \cdot J_A + W_S \cdot J_S \tag{19}$$

where $J$ denotes the objective function; $W_V$, $W_A$, and $W_S$ denote the weighting factors of speed, acceleration, and sharpness, respectively; and $J_V$, $J_A$, and $J_S$ denote corresponding integral functions.

The speed integral function is given by

$$J_V = \int_{t_0}^{t_p} (V_X(t) - V_{Xa})^2 dt = \underbrace{\int_{t_0}^{t_p} V_X^2(t) dt}_{J_{V'}} \underbrace{- 2 \cdot V_{Xa} \cdot \int_{t_0}^{t_p} V_X(t) dt}_{J_{V''}} + \underbrace{V_{Xa}^2 \cdot \int_{t_0}^{t_p} dt}_{const} \tag{20}$$

where the last member that does not contain the variable $q_i$ (see Equation (A4), Appendix A) is omitted.

The speed integral must correspond to the distance $S_p$, considering Equations (A1), (A3), (A8), and (A10) (Appendix A). For the integral function $J_{V''}$,

$$\int_{t_0}^{t_p} V_X(t) dt = \sum_{i=1}^{n} \left( \int_0^{\Delta T_i} f_b^T d\tau \right) \cdot q_i = \sum_{i=1}^{n} g_{bi}^T \cdot q_i = g_f^T \cdot q_f = g^T \cdot E^T \cdot q_f = g^T \cdot E^T \cdot M_q \cdot q \tag{21}$$

where $g = (g_{b1}, g_{b2}, \ldots, g_{bn})^T$, $E = (E_4, E_4, \ldots, E_4)^T$, $E_4$ denotes the identity matrix of the dimension $4 \times 4$, and $M_q$ denotes the transition matrix from the vector $q$ of degrees of freedom to the vector $q_f$ of repeating degrees of freedom of all finite elements (FE). Note that in adjacent FE, the values of the nodes on the right and left are repeated in the vector $q_f$ (e.g., $q_{3i}$ and $q_{4i}$ are equal to $q_{1(i+1)}$ and $q_{2(i+1)}$, respectively). Therefore, the excess degrees of freedom must be reduced by grouping node values instead of FE. That is, $q_f = M_q \cdot q$.

Now consider the integral of the square of speed $V^2(t)$, considering Equation (A11) (Appendix A):

$$J_{V'} = \int_{t_0}^{t_p} V_X^2(t) dt = \sum_{i=1}^{n} q_i^T \cdot D_{bi} \cdot q_i \tag{22}$$

In vector-matrix form, Equation (22) can be written as

$$J_{V'} = \underbrace{\begin{pmatrix} q_1 \\ q_2 \\ \vdots \\ q_n \end{pmatrix}^T}_{q_f} \cdot \underbrace{\begin{pmatrix} D_{b1} & Z_4 & Z_4 & Z_4 \\ Z_4 & D_{b2} & Z_4 & Z_4 \\ Z_4 & Z_4 & \ddots & \vdots \\ Z_4 & Z_4 & \cdots & D_{bn} \end{pmatrix}}_{D} \cdot \underbrace{\begin{pmatrix} q_1 \\ q_2 \\ \vdots \\ q_n \end{pmatrix}}_{q_f} = q_f^T \cdot D \cdot q_f \tag{23}$$

where $Z_4$ = zero matrix ($4 \times 4$).

The integral functions of the acceleration and sharpness are similar to Equation (22). Considering Equation (23) and Equations (A12) and (A13) (Appendix A), they yield

$$J_A = \int_{t_0}^{t_p} \left( \frac{dV_X(t)}{dt} \right)^2 dt = \sum_{i=1}^{n} q_i^T \cdot G_{bi} \cdot q_i = q_f^T \cdot G \cdot q_f \tag{24}$$

$$J_S = \int_{t_0}^{t_p} \left( \frac{d^2 V_X(t)}{dt^2} \right)^2 dt = \sum_{i=1}^{n} q_i^T \cdot K_{bi} \cdot q_i = q_f^T \cdot K \cdot q_f \tag{25}$$

where $G$ and $K$ denote the matrices formed in the same format as $D$ in Equation (23).

By leaving only the members containing the variables, the problem becomes equivalent to quadratic optimization. By substituting Equations (20)–(25), the objective function of Equation (19) becomes

$$\min_{V_X} J = q^T \cdot H \cdot q + 2 \cdot L \cdot q \tag{26}$$

where $H$ denotes the equivalent of Hessian for combined speed, acceleration, and sharpness factors, and $L$ denotes the vector reflecting the average level of speed.

### 4.3. Constraints

#### 4.3.1. Lane Change-Related Constraint

The lane change is the first phase depicted in Figure 1a. The maneuver starts from a position that satisfies the condition of avoiding a blind spot and determines the length of the impeding vehicle (i.e., extremely close to the dashed marking line). The main requirement is that the full departure in the opposite lane should be completed before reaching the rear of the impeding vehicle. This approach ensures maximum security, especially before outrunning a long impeding vehicle, since its lateral behavior (possible swinging of a semitrailer) is unpredictable. Additionally, the tracking of the opposing vehicle's position and speed is retained until the departure to the opposite lane. In an extreme case, a transverse movement may be allowed, where the minimum safe side distance between the passing and impeding vehicles is provided.

The time required for the passing vehicle to pass from State 1 to 2 (Figure 1a) is $t_t$ (Figure 3). Then, the longitudinal component of the path is

$$X_{pb} = \int_{t_0}^{t_t} V_X(t) dt \tag{27}$$

The condition for achieving State 2 (Figure 1a) is

$$X_{pb} \leq d_0 + X_{ib} \tag{28}$$

The time $t_t$ can be determined iteratively (Figure 3) after the distribution of the longitudinal speed, based on Equations (27) and (28), considering $X_{pb} = X_p(t_t)$. According to a possible slight decrease in the impeding vehicle speed, $X_{pb}$ may correspond to a small longitudinal gap between the front of the passing vehicle and the rear of the impeding vehicle. Thus,

$$X_p(t_t) - d_0 - \left( V_{p0} + v_{i0} \right) \cdot t_t \leq 0 \tag{29}$$

The path's transverse component is

$$Y_{pb} = \int_{t_0}^{t_t} V_Y(t) dt \tag{30}$$

where $V_Y(t)$ is represented similar to Equation (21).

#### 4.3.2. Location in Opposite Lane Constraint

The upper limit of the passing vehicle transversal movement $Y_{po}$ may correspond to its position in the middle of the opposite lane. Possible deflections of this position are restricted by safe clearance

to the road edge (Figure 1b). A lower limit of this clearance is the minimum safe distance between passing and impeding vehicles. Therefore, for $X_{po}$ and $Y_{po}$,

$$X_{po} = \int_{t_t}^{t_c} V_X(t)dt \text{ and } Y_{po} = \int_{t_t}^{t_c} V_Y(t)dt = 0 \tag{31}$$

where $t_c$ can be found iteratively according to the condition when the passing and impeding vehicles are abreast at the critical point (State 3, Figure 1b), considering $X_p(t_c) = X_{pb} + X_{po}$ (Point *e*, Figure 3).

According to a possible slight increase in the impeding vehicle speed, the passing vehicle's front can be a little ahead of the front of the impeding vehicle. Thus,

$$X_p(t_c) - (d_0 + L_i) - \left(V_{p0} + v_{i0}\right) \cdot t_c \geq 0 \tag{32}$$

### 4.3.3. Maneuver Completion

The lane change planning is similar to the first phase. Thus,

$$X_{pf} = \int_{t_c}^{t_p} V_X(t)dt \text{ and } Y_{pf} = \int_{t_c}^{t_p} V_Y(t)dt \tag{33}$$

### 4.4. Preparing the Reference Trajectories

As a result of the optimization, the components of speeds $(V_X, V_Y)^T$ in global coordinates are determined. Therefore, it is necessary to transfer the speeds to the local coordinates of the passing vehicle $(V_x, V_y)^T$, in order to allow it to consider its maneuvering. Since the yaw angle $\varphi$ is small,

$$\begin{pmatrix} V_X \\ V_Y \end{pmatrix} = \begin{pmatrix} cos(\phi) & -sin(\phi) \\ sin(\phi) & cos(\phi) \end{pmatrix} \cdot \begin{pmatrix} V_x \\ V_y \end{pmatrix} \tag{34}$$

and

$$\begin{pmatrix} V_x \\ V_y \end{pmatrix} = \begin{pmatrix} cos(\phi) & sin(\phi) \\ -sin(\phi) & cos(\phi) \end{pmatrix} \begin{pmatrix} V_X \\ V_Y \end{pmatrix} \approx \begin{pmatrix} 1 & \phi \\ -\phi & 1 \end{pmatrix} \cdot \begin{pmatrix} V_X \\ V_Y \end{pmatrix} \tag{35}$$

The ideal forecast for the yaw angle can be obtained as a tangent to the motion trajectory:

$$\phi = arctg\left(\frac{dY_p}{dX_p}\right) \approx \frac{V_Y}{V_X} \tag{36}$$

Therefore, to track a virtual trajectory, a state vector corresponding to the capabilities of the current measurements may be used: absolute displacements in global coordinates $X_p$, $Y_p$; speeds in local coordinates $V_x$, $V_y$; and the yaw angle $\varphi$. As additional parameters, which can be directly measured on a vehicle, the accelerations that are components of the optimized speed plans reduced to the vehicle local coordinates may be used, as well as the yaw rate, which can be estimated indirectly as $d\varphi/dt$ considering Equation (36).

## 5. Updating the Speed Plan

Each subsequent measurement determines the new position of the linear forecast. As previously mentioned, the influence of the fluctuations in the participants' speed on the forecast reliability during the maneuver should be analyzed to avoid a redundant number of predictions. In Figure 3, the deviations in the proximity of Point *f* are shown, where the threshold values of changes in speeds of the opposing and impeding vehicles are reached. In Figure 7a1, the speed of the impeding vehicle increases in such a way that the linear curve exceeds the upper boundary prior to the moment $t_a$, and the segment *b'k'* slightly goes up (green), along the path curve of the impeding vehicle. Basically, the value of the minimum required distance $d_{fmin}$ depends on the difference between the speeds of

the passing and impeding vehicles, and, thus, will vary with the fluctuations in movement modes of the overtaking participants. However, the changes will not have a significant affect, and therefore, $d_{fmin}$ can be considered constant in the vicinity of Point $t_a$. The determination of $d_{fmin}$ is described elsewhere [24,25]. The bias of the intersection point of the trapezoid diagonals in $z'$ leads to shifting of the optimal Point $f'$ up to the left. The required time $t'_p$ becomes shorter and the needed space $X'_p$ becomes larger. This may be explained by the significant sensitivity of the forecast to the impeding vehicle's speed changes.

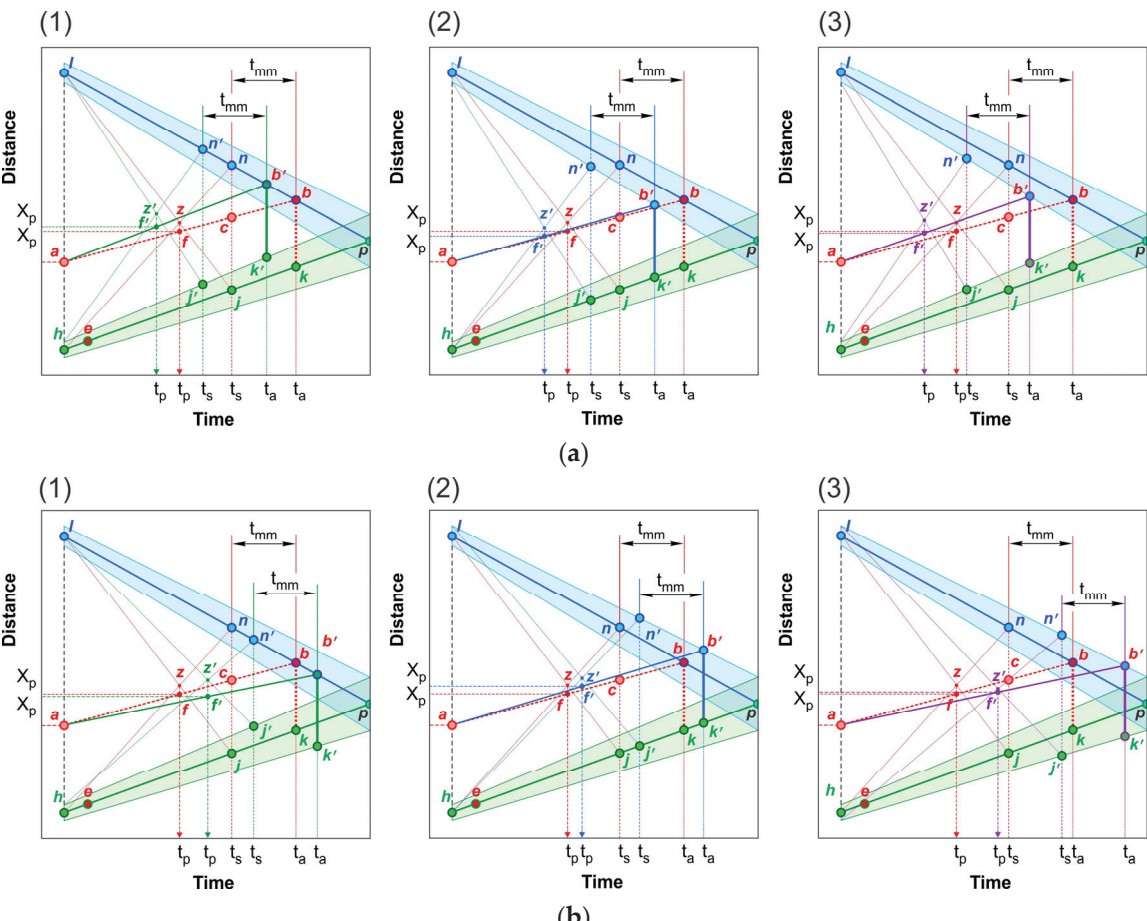

**Figure 7.** Influence of threshold values of speed fluctuations on the prediction reliability: (**a**) Increase in speeds of opposing and impeding vehicles (Cases 1-3); (**b**) decrease in speeds of opposing and impeding vehicles (Cases 1-3).

In the case of Figure 7a2, the opposing vehicle speed increases and the upper boundary limit is violated. In this case, the inclination angles of the segments *ab* and *ab'* are practically the same. However, even though the required time has decreased $t'_p < t_p$, unlike the previous case, the required space $X'_p$ decreases due to the larger space needed for the opposing vehicle. The most critical case is when the speed fluctuations of both the opposing and impeding vehicles reach the threshold boundaries simultaneously (Figure 7a3). The displacement of the minimum space segment $d_{fmin} + L_p$ for completing the maneuver can be so significant that the time $t'_s$ approaches the preset time $t_p$, and the margin $t_{mm}$ in relation to the minimum performance mode will not be provided. Point $t'_p$ is located the furthest from Point $t_p$, even though the space required for the maneuver may remain almost unchanged $X'_p \approx X_p$. Similarly, a decrease in the speed of opposing and impeding vehicles will give a lower limit of time fluctuations $f_L$ (Figure 3). However, such decreases are not dangerous, and it makes sense to only recalculate the forecast to save energy and increase the movement stability.

Based on the described scheme, it is possible to determine the allowable level of deviations, at which the margin of minimum safety is kept without the necessary recalculation.

Therefore, for the case depicted in Figure 7b1, the diminished impeding vehicle speed means that a shorter distance and longer period are needed: $X'_p < X_p$ and $t'_p > t_p$. For the case in Figure 7b2, the diminished opposing vehicle speed demonstrates the need for a longer distance and time because of the reduced space $X_o$ for the opposing vehicle: $X'_p > X_p$ and $t'_p > t_p$. For the case in Figure 7b3, the diminished speeds of both opposing and impeding vehicles move Point $z'$ quite far from $z$, providing a longer time in almost the same space: $X'_p \approx X_p$ and $t'_p > t_p$. This may cause the double margin time $t_{mm}$ with unreasonable energy consumption. Therefore, the passing vehicle speed mode may be reduced.

Other possible combinations, where one of the vehicles increases its speed and the other decreases its speed approximately simultaneously, give solutions that are not superior in nature to the changes discussed in Figure 7. Note that measurements of $d_n$ are available prior to the moment of alignment with the rear of the impeding vehicle, after which the last $d_n$ value may be fixed, and measurements of $D_n$ can be carried out until the critical point. Recalculation after the critical point is possible if the next impeding vehicle appears in the lane, which does not provide a proper pocket or harshly reduces its speed.

Obviously, even in the automatic mode of maneuver execution, deviations of the passing vehicle speed are possible due to the influence of various random factors. However, within the thresholds set by $f_U, f_L$ (Figure 3), forecast recalculation is not required. Therefore, the autonomous control system must adjust the speed mode of the passing vehicle not only according to measurement changes, but also considering the matching with its own reference curve (*of* in Figure 3).

The condition under which the specified safety level is retained and the forecast does not require recalculation is

$$\begin{cases} t'_a > t_p + t_{mm} + t_{pr} + t_{un}, \; if \; t'_a < t_a \\ t'_a > t_a + t_{mm} - \left(t_{pr} + t_{un}\right), \; if \; t'_a > t_a \end{cases} \tag{37}$$

where $t'_a$ denotes the instantaneous value of the hypothetical accident time compatible with predefined $t_a$; $t_{pr} = \Delta t_{pr} + p \cdot \Delta t_m = (m+p) \cdot \Delta t_m$, where $p$ denotes the number of spare measurement cycles, by default, $p = 2$; and $t_{un}$ denotes the unaccounted time expenses (e.g., engine transition mode and control delay).

Hence, $t'_a$ can be recalculated using Equation (13) for every *n*-th measurement at $t_n$ as follows:

$$t'_a = \frac{D_n - d_n - L_i - d_{fmin} - L_p}{v_{in} - v_{on}} + t_n \tag{38}$$

As can be seen, the expression does not contain any absolute value, except for vehicles' lengths, and only uses relative measurements, making it independent of the passing vehicle state parameters, including variance regarding its reference curve.

## 6. Overtaking Scenario modeling

### 6.1. Vehicle Model Description

Consider the simple linear single track ("bicycle") vehicle model in the standard form of state-space:

$$dx/dt = A \cdot x + B \cdot u \; \text{and} \; y = C \cdot x + D \cdot u \tag{39}$$

where $x$ denotes the state vector; $y$ denotes the output vector; $u$ denotes the control vector; and $A$, $B$, $C$, and $D$ are matrices.

Suppose that the vehicle maintains its longitudinal speed $V_x$ from Equation (34), and the state vector $x$ only contains parameters for the front wheel steering control $u = \Theta_f$. Therefore, $x = (V_y, Y, \omega,$

$\varphi)^T$, where $Y$ denotes the lateral displacement in global coordinates, and $\omega = d\varphi/dt$ denotes the yaw rate. Other parameters are denoted above. Provided $D = 0$, the matrices $A$, $B$, and $C$ can be derived as

$$A = \begin{pmatrix} -\frac{k_f+k_r}{m\cdot V_x} & 0 & -V_x - \frac{k_f\cdot x_f+k_r\cdot x_r}{m\cdot V_x} & 0 \\ 1 & 0 & 0 & V_x \\ -\frac{k_f\cdot x_f+k_r\cdot x_r}{I\cdot V_x} & 0 & -\frac{k_f\cdot x_f^2+k_r\cdot x_r^2}{I\cdot V_x} & 0 \\ 0 & 0 & 1 & 0 \end{pmatrix}, \ B = \begin{pmatrix} \frac{k_f}{m} \\ 0 \\ \frac{x_f\cdot k_f}{I} \\ 0 \end{pmatrix}, \ C = \begin{pmatrix} 0 & 0 \\ 1 & 0 \\ 0 & 0 \\ 0 & 1 \end{pmatrix}^T \quad (40)$$

where $m$ and $I$ denote the vehicle mass and inertia, respectively; $k_f$ and $k_r$ denote the front and rear tires' side stiffness, respectively; and $x_f$ and $x_r$ denote the local longitudinal coordinates of the front and rear tire spots, correspondingly. Therefore, the output variables for reference tracking are $y = (Y, \varphi)^T$, which, in the real world, can be measured using a camera and sensors.

*6.2. Adaptive Model Predictive Control Tracking Optimization Problem*

According to the tracking problem, the control parameters should provide the values closest to the reference signals. Therefore, the cost function for the Adaptive Model Predictive Control (AMPC) [26] controller may be composed of the minimization of the sum of the squared errors, as follows:

$$\min_u J(z_k) = \rho_\varepsilon \cdot \varepsilon_k^2 + \sum_{i=0}^{p-1} \left( e_{y,k+i}^T \cdot Q_y \cdot e_{y,k+i} + e_{u,k+i}^T \cdot Q_u \cdot e_{u,k+i} + e_{\Delta u,k+i}^T \cdot Q_{\Delta u} \cdot e_{\Delta u,k+i} \right) \quad (41)$$

subject to $e_{y,k+i} = y_{k+i+1|k}^* - y_{k+i+1|k}, \ e_{u,k+i} = u_{k+i|k}^* - u_{k+i|k}, \ e_{\Delta u,k+i} = u_{k+i|k} - u_{k+i-1|k}$ (42)

where $Q_y$, $Q_u$, and $Q_{\Delta u}$ denote positive semi-defined weight matrices; $y_{k+i+1|k}^*$ denotes the *Plant* output reference signals at the *i*th prediction horizon step; $y_{k+i+1|k}$ denotes the *Plant* outputs at the *i*th prediction horizon step; $u_{k+i|k}^*$ denotes the *Plant* target reference signals at the *i*th prediction horizon step; $u_{k+i|k}$ denotes the *Plant* inputs (manipulated variables) at the *i*th prediction horizon step; $z_k = (u_{k|k}^T, u_{k+1|k}^T, \cdots u_{k+p-1|k}^T, \varepsilon_k)$ denotes the solution; $\varepsilon_k$ denotes the scalar dimensionless slack variable used for constraint softening at control interval $k$; $\rho_\varepsilon$ denotes the constraint violation penalty weight; $k$ denotes the current control interval; and $p$ denotes the prediction horizon (number of intervals).

The system of constraints is written as

$$\begin{cases} y_{j,min(i)} - \varepsilon_k \cdot h_{j,min(i)}^{(y)} \le y_{j,k+i|k} \le y_{j,max(i)} + \varepsilon_k \cdot h_{j,max(i)}^{(y)}, & i = 1\ldots p, \ i = 1\ldots n_y \\ u_{j,min(i)} - \varepsilon_k \cdot h_{j,min(i)}^{(u)} \le u_{j,k+i-1|k} \le u_{j,max(i)} + \varepsilon_k \cdot h_{j,max(i)}^{(u)}, & i = 1\ldots p, \ i = 1\ldots n_u \\ \Delta u_{j,min(i)} - \varepsilon_k \cdot h_{j,min(i)}^{(\Delta u)} \le \Delta u_{j,k+i-1|k} \le \Delta u_{j,max(i)} + \varepsilon_k \cdot h_{j,max(i)}^{(\Delta u)}, & i = 1\ldots p, \ i = 1\ldots n_{\Delta u} \end{cases} \quad (43)$$

*where $y_{j,min(i)}$ and $y_{j,max(i)}$ denote the minimum and maximum values of the jth output at the ith prediction horizon step, respectively; $u_{j,min(i)}$ and $u_{j,max(i)}$ denote the minimum and maximum values of the jth input at the ith prediction horizon step, respectively; $\Delta u_{j,min(i)}$ and $\Delta u_{j,max(i)}$ denote the minimum and maximum values of the jth input rate at the ith prediction horizon step, respectively; $h^{(y)}_{j,min(i)}$ and $h^{(y)}_{j,max(i)}$ denote the minimum and maximum values of the jth output's hard constraints at the ith prediction horizon step, respectively; $h^{(u)}_{j,min(i)}$ and $h^{(u)}_{j,max(i)}$ denote the minimum and maximum values of the jth input's hard constraints at the ith prediction horizon step, respectively; $h^{(\Delta u)}_{j,min(i)}$ and $h^{(\Delta u)}_{j,max(i)}$ denote the minimum and maximum values of the jth input rates' hard constraints at the ith prediction horizon step, respectively; $n_y$ denotes the number of output parameters; $n_u$ denotes the number of input parameters; and $n_{\Delta u}$ denotes the number of input rate parameters.6.3. Simulink Model*

A simplified Simulink model (Figure 8) that implements a virtual overtaking scenario on a two-lane highway has been developed. The main block 1 (*SUV Plant*) calculates the state vector of the vehicle's continuous dynamic model, which presents a real vehicle and its sensor system measurements.

Block 2 (*SUV Model*) calculates the vector of discrete states of the vehicle dynamic model, updating the necessary matrices and vectors at each time step. For simplicity, the same bicycle vehicle model is used as a *Plant* and *Model*. Block 3 (*MPC*) implements the Adaptive Model Predictive Controller, which calculates the optimal control values (steering angle), based on minimizing the sum of the square of the differences measured and predicted parameters: lateral displacement *Y* and yaw angle *Phi*, which is extracted from the *Plant* state vector by block 5 (*MO Extractor*). The vectors of the reference tracks *Ref*, being the desirable values of the vehicle model state parameters, are stored in the memory after optimization for reading at the corresponding time step. Block *4* (*Conditions*) sets the values for the vehicle's local longitudinal speed $V_x$ and the desired reference values $Ref = (Y, Phi)^T$ at the current time. Block *6* (*Result*) accumulates the calculated outputs. The model does not comprise external disturbances and measurement noise.

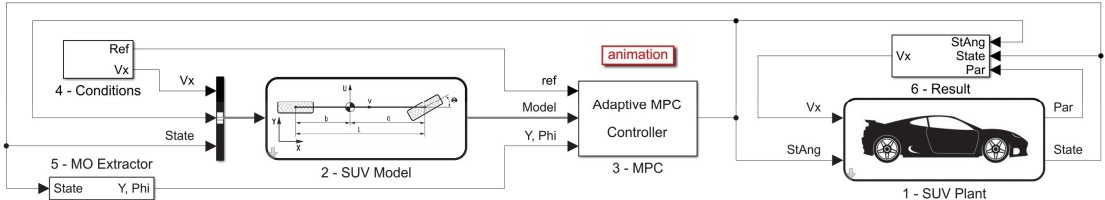

**Figure 8.** Simulink model of overtaking scenario execution.

## 7. Application

The purpose of this application is to achieve stable control and ensure that the *Plant* state parameters fit those generated by the proposed methodology. The application involved setting initial conditions data, defining the parameters of the AMPC controller, and determining the desired reference tracks for speed, acceleration, and displacement in the global coordinates.

### 7.1. Initial Conditions Data

The Matlab/Simulink example for simulating overtaking was used [26]. According to the measurements, at time $T_{-1} = -0.1$ s, the initial data vector is formed as follows:

$$(D_{(-1)}, V_{o(-1)}, d_{(-1)}, V_{i(-1)}, L_{i(-1)}, X_{p(-1)}, V_{p(-1)}) = (480, 70, 35, 65, 22.5, 0, 70),$$

where the linear dimensions are given in *m* and speeds in *km/h*. Using the technique described in [22], for the case of ideal motion conditions, the necessary values yield $t_{min} = 7.79$ s and $S_{fmin} = 25$ m. The minimum time margin was set as $t_{mm} = 1$ s. Substituting these values into Equations (12)–(18) and (27)–(33) gives the following rational values (Figure 9a,b): overtaking global longitudinal projection $X_p = 250$ m, overtaking time $t_p = 8.9 \approx 9$ s, bypass time during lane change $t_t = 5.1$ s, and time to the critical Point $t_c = 6.9$ s.

### 7.2. Parameters of the AMPC Controller

The sampling time = *0.1* s, prediction horizon = *10* s, and control horizon = *2* s. The plant model has four states with two measured outputs. Weights: manipulated variable (steering angle) = *0* and manipulated variable rate (steering angle rate) = *0.1*; output variables: lateral displacement = *0.8* and yaw angle = *0.1*; constraints: $-0.2 \leq$ steering angle (rad) $\leq 0.2$, $-0.2 \leq$ steering angle (rad/s) $\leq 0.2$, $0 \leq$ lateral displacement (m) $\leq 3.6$, and $-0.1 \leq$ yaw angle (rad) $\leq 0.1$.

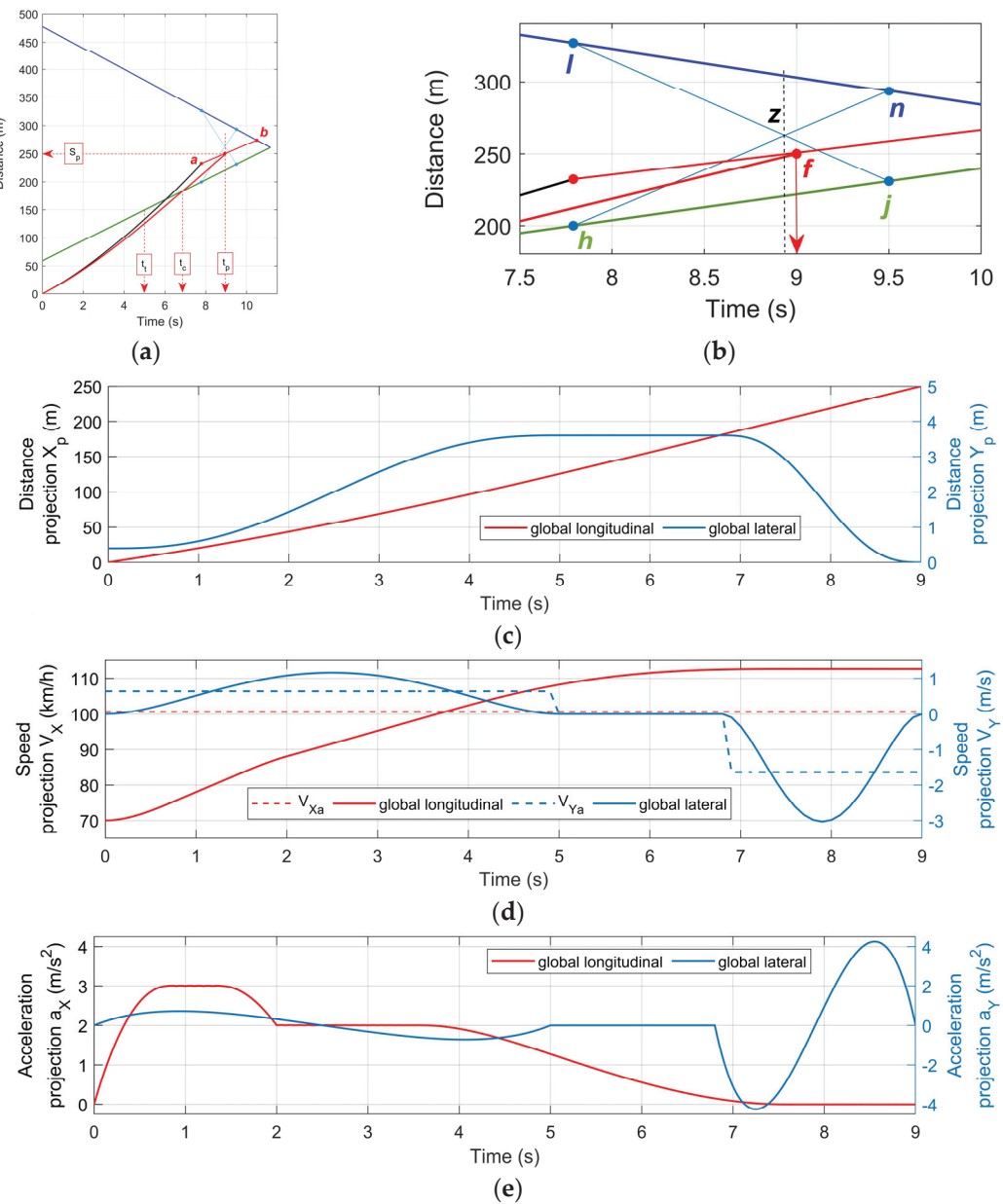

**Figure 9.** Planning reference tracks for state parameters: (**a**) Vehicle path's prognosis; (**b**) definition of point $t_p$; (**c**) predicted passing vehicle's global displacements; (**d**) plan of global velocities; (**e**) plan of global accelerations.

### 7.3. Reference Speeds, Accelerations, and Displacements

Using the proposed optimization model, the desired reference tracks for speed, acceleration, and displacement in the global coordinates were determined, using the time grid with the increment of *0.1* s (Figure 9c–e). Note that the setting of linear constraints in Equation (40) should be consistent with the thresholds of the vehicle performance set by the distribution of acceleration upon speed (Figure 4) for the current conditions. This means that each speed value to be optimized in a time grid node is tied to the maximum possible acceleration at this speed, which creates non-linear constraints. In this regard, for each set of speed and acceleration thresholds in the optimization process, some rationale is needed, as previously described. Another difficult point before optimization is setting the final values of the speed and acceleration, since they significantly affect the trend of the entire speed plan. The speed value in the last node close to the average speed $V_{Xa}$ (Figure 9d) may lead to the appearance of such a peak near the critical point, when the longitudinal accelerations in the phase of maneuver completion

are negative and larger than the absolute value of *0.5 m/s²*. That would mean the use of service braking and activation of the vehicle's working brake system. From the point of view of ensuring the maximum vehicle stability during the lane change, it is undesirable to use the tire longitudinal force values close to those which may considerably reduce the tire's lateral adhesion. In connection with the foregoing, it may be recommended to focus on the value of the speed $V_{pf}$ at which the distribution of the speed plan requires decelerations, provided only by limiting the engine power consumption. In this case, the selected value $V_{pf}$ ensures maneuver completion with acceleration close to zero.

As can be seen, the combination of the plan for longitudinal speeds and accelerations fits the performance limitations well in Figure 4. At the same time, the projection curve of the overtaking path on the global *X* axis clearly corresponds to *250* m (Figure 9d). This, when copied to Figure 9a's curve, shows that in the initial phase, the vehicle uses a potential close to the upper limit (black line). The next important point is the conditionality of the weighting factors in the optimization of Equation (19). Note that the ratio of weight coefficients significantly changes the optimization picture in connection with the change of priorities. Increasing the $W_V$ coefficient very much reduces the speed consumption, but significantly increases the need for acceleration at the beginning of the maneuver. The increase of the $W_A$ coefficient reduces the cumulative consumption of acceleration, but does not provide smoothness in the boundary zones of the speed plan, and the peak speed value rises. Increasing the $W_S$ coefficient distributes speeds evenly over time.

Therefore, in the current case of optimizing the longitudinal plan, the stable engine's performance is the most important, minimizing abrupt transitions in its control; respectively, the values for the entire overtaking maneuver are chosen: $W_V = 0.2$, $W_A = 0.2$, and $W_S = 0.6$. In the distribution of transverse speeds of the bypass phase, the priority is divided between the control smoothness and the cumulative acceleration intake: $W_V = 0.2$, $W_A = 0.4$, and $W_S = 0.4$. In the final phase, due to the lane change at high speeds, the main priority is focused on reducing the lateral accelerations, respectively: $W_V = 0.1$, $W_A = 0.6$, and $W_S = 0.3$. It is obvious, however, that priorities may vary, depending on the situation.

It should also be noted that the value of the vehicle's initial lateral position does not correspond to the lane center, but is offset by *0.4* m from the dashed line to ensure the conditions previously described.

Figure 10 shows the overtaking results by predicting the lateral offset and yaw angle. As can be seen in Figure 10a, at the 9th second of overtaking, the trajectory longitudinal component practically corresponds to the pre-set one, with a final value of $X_p = 250 \approx 249.4$ m, and the transverse component $Y_p$ is strictly within *3.6* m, but has a residual of *0.23* m at the time $t_p = 9$ s. At this moment, the passing vehicle is almost in the middle of its lane and continues stable movement, i.e., the situation is uncritical. The AMPC controller calculates the discrete control signal based on the information on the previous value and reference tracks. However, it is almost impossible to avoid tracking delay completely. The same effect can be observed in relation to the lateral speed $V_Y$ (Figure 10b), which coincides in terms of shape and values with the initial one in Figure 9d, but lags a bit in time.

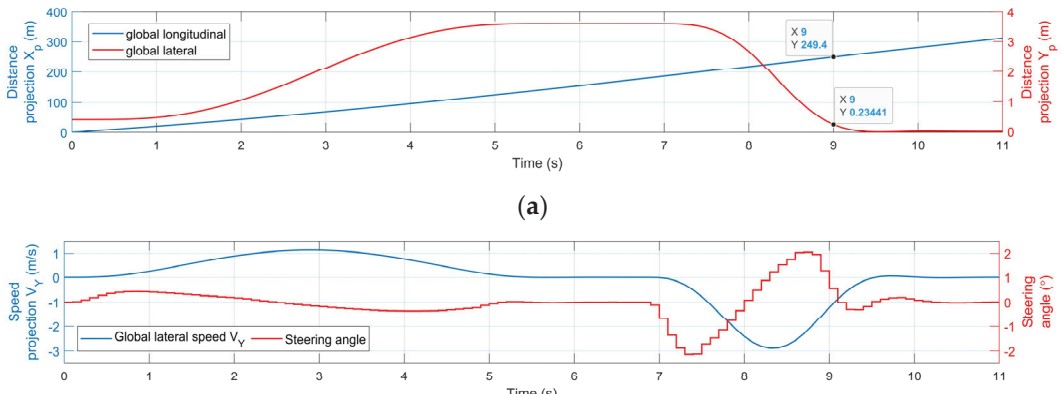

**Figure 10.** Simulation results of vehicle steering control prognosis during overtaking: (**a**) Global displacements; (**b**) steering control and lateral speed projection.

In general, by using the vehicle refined models of *Plant* and *Model* (Figure 8) and including a larger number of state parameters for tracking and other measures, the model convergence can be improved.

## 8. Concluding Remarks

This article has presented a methodology for distributing the speed in the longitudinal and lateral directions when a vehicle is overtaking on two-lane highways in an automated mode. An advantage of the kinematic technique used in the model is its ability to predict both speed and acceleration references, providing subsequent tracking control based on sensor measurements. In addition, this technique can be successfully used as a component of the model predictive control for generating reference trajectories. Based on this study, the following comments are offered.

In this study, the 'kinematic model' was not exactly used in the sense of a vehicle model, but the output kinematic parameters of the vehicle motion were used to predict both the trajectory and the overtaking mode, considering the speed distribution directly and acceleration distribution indirectly, as part of the FE speed model. In this regard, we followed the well-established trend of planning motion paths with various curves and graphs, as presented in the literature [17]. However, we obtained trajectories as integrals of the velocity plans. Unlike other studies, the proposed approach focused on using an FE grid that simultaneously ensured a high accuracy and compliance with the real capabilities of the vehicle engine-drive system. In particular, the interaction between possible speeds and the respective accelerations was considered.

In the quadratic optimization model for the speed-plan distribution, three measures were considered (speed, acceleration, and sharpness). The two parameters ($t_p$ and $X_p$) characterize the average speed $V_{xa}$ and are obviously not enough, because many curves integrable in the interval ($t_0$, $t_p$) can give the same $X_p$. In addition, the vehicle's ability to increase speed is a function of the speed and depends on the vehicle's characteristics (i.e., acceleration is a 3D surface as a function of vehicle speed and throttle activation level). Therefore, in this study, it was assumed that the speed nodes are interconnected by curves over time, differentiable at least twice (i.e., providing smoothness). Then, the derivatives at the nodes which reflect the slope (acceleration) and the curvature (sharpness) were included as members of the objective function, along with their weights. Since this is an FE model of the curve, no other parameters except for the nodal $V_x$, $dV_x/dt$, and $d^2V_x/dt^2$ can be included, because only the nodes need to be distributed, consistent with the vehicle technical features.

The general idea of using the overtaking heuristic algorithm is to guarantee maneuver completion, prior to the maneuver beginning, with at least the minimum safety margin. The algorithm emanates from individual vehicle operational properties. This allows the maximum and minimum performances to be predicted for given initial conditions. Therefore, considering possible changes due to random factors, for example, the algorithm will provide a decision to only pass when maneuver safety is ensured. In addition, the algorithm can foresee the situation when a vehicle must have a power margin, and whether maneuver completion requires additional acceleration to prevent a possible collision. Note that cancelling the overtaking maneuver is possible. However, within the framework of the presented algorithm, cancellation could happen in occasional situations (e.g., sudden swinging of the impeding vehicle). In this case, a switch to another algorithm is needed and this should be investigated in the future.

Determination of the required overtaking time is based on the safety margin and adequate engine power. These criteria allow possible adjustments of the power consumption and safety time, depending on the situation changes and priorities. A simple, but quite effective, technique was proposed in the heuristic algorithm for finding one rational curve from an infinite number of possible realizations. This is the reason why two additional derivatives (acceleration and sharpness) were used for finding the best speed distribution. All these measures aim at ensuring strict vehicle power stability and consequently, safety. In addition, by considering the lateral acceleration and yaw rate (indirectly) in the steering control prognosis, motion instability and possible sideslip can be prevented. Therefore, maneuver safety has been adequately incorporated in the proposed framework.

In this study, the roadway is assumed to be straight, with ideal surface friction and no external forces (e.g., gusts of wind). In addition, the road is assumed to have very gentle vertical curvature. These assumptions are necessary, since sensor measurements can be particularly affected by the slope and direction of the road, being unable to measure the position and speed of the opposing vehicles in various circumstances. Fortunately, this does not represent a limitation of the model, since overtaking maneuvers on two-lane highways are not permitted on sharp vertical curves because they require very long vertical curves that are expensive to construct [27,28].

Further research can be conducted to improve the algorithm for finding the optimal speed distribution for overtaking. Areas of focus may include: (1) influence of the final speed of the maneuver on the nature of the optimal speed plan; (2) influence of the weight coefficients on the speed plan; and (3) modeling of obstacle avoidance in autonomous overtaking. By grouping the longitudinal and lateral components of the reference trajectories, the kinematic technique can be used to simulate the obstacle-avoidance trace for autonomous vehicles. In addition, a sensitivity analysis of the proposed framework can be conducted to understand the impact of certain problem parameters on the overall results, including the effect of uncertainty.

**Author Contributions:** Conceptualization, S.M.E. and M.D.; methodology, M.D. and S.M.E.; software, M.D.; validation, S.M.E.; formal analysis, S.M.E.; investigation, M.D.; resources, S.M.E.; data curation, S.M.E.; writing-original draft preparation, M.D. and S.M.E.; writing-review and editing, S.M.E. and M.D.; visualization, M.D.; supervision, S.M.E.; project administration, S.M.E.; funding acquisition, S.M.E. All authors have read and agreed to the published version of the manuscript.

**Funding:** This research was sponsored by the Natural Sciences and Engineering Research Council of Canada (NSERC).

**Acknowledgments:** The authors are grateful to three anonymous reviewers for their thorough and most helpful comments.

**Conflicts of Interest:** The authors declare no conflicts of interest. The funders had no role in the design of the study; in the collection, analyses, or interpretation of data; in the writing of the manuscript; or in the decision to publish the results.

## Abbreviations

The following abbreviations are used in this manuscript:

| | |
|---|---|
| NMPC | Model Predictive Control |
| RRT | Rapid Random Tree |
| GPS | Global Positioning System |
| LQP | Linear Quadratic Programming |
| AMPC | Adaptive Model Predictive Control |
| SUV | Single Unit Vehicle |
| MO | Measured Outputs |
| FE | Finite Element |

## Appendix A : Representing the Speed Function by Finite Elements

Suppose that the speed of the passing vehicle within the time interval $(t_0, t_p)$ varies along the *X*-coordinate of the road segment $0-X_p$, according to the law $V_X(t)$. Then, for a grid of **n** time intervals $(t_0, t_1, t_2, \ldots, t_n)$,

$$X_p = \int_{t_0}^{t_p} V_X(t)dt = \sum_{i=1}^{n} \int_{t_{i-1}}^{t_i} V_X(t)dt = \sum_{i=1}^{n} \int_{0}^{\Delta T_i} V_{Xi}(\tau, \Delta T_i)d\tau \tag{A1}$$

where $\Delta T_i$ is the time interval $(t_i - t_{i-1})$, which is generally variable.

Using the FE method for the piecewise representation of the speed function of each interval, for the **i**-th time segment $(t_{i-1}, t_i)$,

$$V_{Xi}(\tau, \Delta T_i) = \sum_{k=1}^{4} q_{ki} \cdot f_{\tau k}(\tau, \Delta T_i) \tag{A2}$$

where $\tau \in [0, \Delta T_i]$ is the FE local time; $q_{1i}, q_{2i}, q_{3i}$, and $q_{4i}$ represent impact coefficients, where $q_{1i}$ and $q_{3i}$ are speeds at FE nodes and $q_{2i}$ and $q_{4i}$ are accelerations (derivatives) at the corresponding nodes; and $f_{\tau 1}, f_{\tau 2}, f_{\tau 3}$, and $f_{\tau 4}$ are basis functions.

Thus, using matrix notation,

$$V_{Xi}(\tau, \Delta T_i) = \boldsymbol{f}_b^T(\tau, \Delta T_i) \cdot \boldsymbol{q}_i = \boldsymbol{f}_b^T \cdot \boldsymbol{q}_i \tag{A3}$$

where

$$\boldsymbol{f}_b = \begin{pmatrix} f_{\tau 1} & f_{\tau 2} & f_{\tau 3} & f_{\tau 4} \end{pmatrix}^T, \ \boldsymbol{q}_i = \begin{pmatrix} q_{1i} & q_{2i} & q_{3i} & q_{4i} \end{pmatrix}^T \tag{A4}$$

The normalized basis function $f_\xi$ for an FE of a unitary length ($\Delta T = 1$) is based on the cubic polynomial with two degrees of freedom at a node, providing smoothness and continuous differentiability, as follows:

$$\boldsymbol{f}_\xi = \begin{pmatrix} f_{\xi 1} \\ f_{\xi 2} \\ f_{\xi 3} \\ f_{\xi 4} \end{pmatrix} = \begin{pmatrix} (2 \cdot \xi + 1) \cdot (\xi - 1)^2 \\ \xi \cdot (\xi - 1)^2 \\ -\xi^2 \cdot (2 \cdot \xi - 3) \\ \xi^2 \cdot (\xi - 1) \end{pmatrix}, \ \boldsymbol{l}_{\Delta T} = diag \begin{pmatrix} 1 \\ \Delta T \\ 1 \\ \Delta T \end{pmatrix} \tag{A5}$$

where $\xi \in [0, 1]$ is the normalized coordinate.

Let $\tau = \xi \cdot \Delta T$. Then, the transition between the absolute and normalized basis functions is given by

$$\boldsymbol{f}_b = \boldsymbol{f}_b(\tau, \Delta T) = \boldsymbol{f}_b(\Delta T \cdot \xi, \Delta T) = \boldsymbol{l}_{\Delta T} \cdot \boldsymbol{f}_\xi. \tag{A6}$$

Considering the basis functions defined only within the FE interval, the relation for the entire speed can be written as

$$V_X(t) = \sum_{i=1}^n \boldsymbol{f}_b^T(\tau, \Delta T_i) \cdot \boldsymbol{q}_i = \sum_{i=1}^n \boldsymbol{f}_{bi}^T \cdot \boldsymbol{q}_i. \tag{A7}$$

Since the function also uses (up to second) derivatives of FE basis functions, the first and second derivatives are obtained, respectively, as

$$\frac{d\boldsymbol{f}_b}{d\tau} = \frac{\boldsymbol{l}_{\Delta T}}{\Delta T} \cdot \frac{d\boldsymbol{f}_\xi}{d\xi} \tag{A8}$$

$$\frac{d^2\boldsymbol{f}_b}{d\tau^2} = \frac{\boldsymbol{l}_{\Delta T}}{\Delta T^2} \cdot \frac{d^2\boldsymbol{f}_\xi}{d\xi^2} \tag{A9}$$

Consider the formation of common integrals, replacing the differential $d\tau = d\xi \cdot \Delta T$ and thresholds. Since $\boldsymbol{q}_i$ does not depend on $\tau$, only the basis functions (Equation (A5)) are integrated.

$$\boldsymbol{g}_b = \int_0^{\Delta T} \boldsymbol{f}_b(\tau, \Delta T) d\tau = \Delta T \cdot \boldsymbol{l}_{\Delta T} \cdot \int_0^1 \boldsymbol{f}_\xi d\xi \tag{A10}$$

$$D_b = \int_0^{\Delta T} \boldsymbol{f}_b \cdot \boldsymbol{f}_b^T d\tau = \Delta T_i \cdot \boldsymbol{l}_{\Delta T} \cdot \left( \int_0^1 \boldsymbol{f}_\xi \cdot \boldsymbol{f}_\xi^T d\xi \right) \cdot \boldsymbol{l}_{\Delta T}^T \tag{A11}$$

$$G_b = \int_0^{\Delta T} \frac{d\boldsymbol{f}_b}{d\tau} \cdot \frac{d\boldsymbol{f}_b^T}{d\tau} d\tau = \frac{\boldsymbol{l}_{\Delta T}}{\Delta T} \cdot \left( \int_0^1 \frac{d\boldsymbol{f}_\xi}{d\xi} \cdot \frac{d\boldsymbol{f}_\xi^T}{d\xi} d\xi \right) \cdot \boldsymbol{l}_{\Delta T}^T \tag{A12}$$

$$K_b = \int_0^{\Delta T} \frac{d^2\boldsymbol{f}_b}{d\tau^2} \cdot \frac{d^2\boldsymbol{f}_b^T}{d\tau^2} d\tau = \frac{\boldsymbol{l}_{\Delta T}}{\Delta T^3} \cdot \left( \int_0^1 \frac{d^2\boldsymbol{f}_\xi}{d\xi^2} \cdot \frac{d^2\boldsymbol{f}_\xi^T}{d\xi^2} d\xi \right) \cdot \boldsymbol{l}_{\Delta T}^T \tag{A13}$$

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
