# Peer review of "Optimal Speed Plan for the Overtaking of Autonomous Vehicles on Two-Lane Highways"

_infrastructures, doi:10.3390/infrastructures5050044_

Round 1
Reviewer 1 Report
The paper presents a new technique for speed planning in the case of AV overtaking on two-lane highways. The method combines the use of heuristic algorithms with quadratic optimization modeling to ensure a smooth path of the passing vehicle. Overall, this is an excellent paper that tackles an interesting topic that is worthy of investigation. I was very intrigued by the information presented in this paper, as I found it to be both novel and informative. Here a few remarks for improvement.
Major comments:
The authors are encouraged to provide an in-depth justification for their reliance on kinematic model
Page 11 line 364: please provide more explanation on the rational for the variables used in the objective function. It seems to me that the function can be written to include more variables, so why there the speed, acceleration and sharpness the only three measures included.
Is it possible to conduct a sensitivity analysis of the proposed framework to understand the impact of certain problem parameters on the overall results? It would be interesting to explore the uncertainty in the output.
The paper is mathematically extensive, and I must admit that some portions of the formulation was beyond my capacity. However, I am intrigued by some of the assumptions. Looking through the document it was hard to pinpoint them all. I suggest, if the authors can include a discussion of their assumptions in a separate section to allow the readers to better understand the context. One issue I would like to inquire about is how was sensory noise accounted for in the proposed algorithm? Or was it assumed that such noise did not exist. This and similar assumptions should be better documented throughout the entire paper.
Minor comments:
Please remove the last paragraph in the introduction. These roadmaps represent an outdated style of writing. The information on paper organization is hardly relevant and is mostly glossed over by readers. A reviewer mentioned this to me a few years back and I have found that eliminating these roadmaps have greatly improved the flow of my own papers. I suggest that the authors do the same.
Page 4 line 121: “The proposed system assumes a straight horizontal road, ideal friction with the road surface, and absence of additional external forces such as wind” Please consider revising this sentence. For example “a straight horizontal road” should be “straight sections of roads” or “tangents”. Also, “ideal friction with the road surface” should be revise to “ideal surface friction” and remove the word “additional” before external factors
Page 4 line 126: “passing vehicle is ready starting the maneuver” do you mean “passing vehicle is already starting the maneuver” or “passing vehicle is ready to start the maneuver”
Page 4 line 130: “time moment” should be “time” only delete moment.
Author Response
RESPONSE TO COMMENTS REVIEWER 1
Paper Title: Optimal Speed Plan for Overtaking of Autonomous Vehicles on Two-Lane Highways
Authors: Said M. Easa and Maksym Diachuk
Ms. No.: Infrastructures -779954
We would like to thank you for your thorough and most helpful comments that have helped improved the paper. Below are the responses to each comment (in blue).
Comments and Responses
The paper presents a new technique for speed planning in the case of AV overtaking on two-lane highways. The method combines the use of heuristic algorithms with quadratic optimization modeling to ensure a smooth path of the passing vehicle. Overall, this is an excellent paper that tackles an interesting topic that is worthy of investigation. I was very intrigued by the information presented in this paper, as I found it to be both novel and informative. Here a few remarks for improvement.
Response: Thank you so much for your kind and encouraging comments.
Major comments:
The authors are encouraged to provide an in-depth justification for their reliance on kinematic model
Response: In this study, the ‘kinematic model’ was not exactly used in the sense of a vehicle model, but the output kinematic parameters of the vehicle motion were used to predict both the trajectory and the overtaking mode (speed distribution directly and acceleration distribution indirectly as part of the speed FEM-model). In this regard, we followed the 20-year trend of planning motion paths with various curves and graphs as presented in the literature [2, 3]. That is, we used the well-established principle of motion planning, but we obtained trajectories as integrals of the velocity plans. Unlike other studies where up to 5th order curves are used for transitions and vehicle technical characteristics were not explicitly considered, our approach focused on using a finite element grid that simultaneously ensured high accuracy and compliance with the real capabilities of the vehicle engine-drive system. In particular, the interaction between possible speeds and the respective accelerations was considered.
The preceding explanation was added as Item 1 in Concluding Remarks, as follows:
“In this study, the ‘kinematic model’ was not exactly used in the sense of a vehicle model, but the output kinematic parameters of the vehicle motion were used to predict both the trajectory and the overtaking mode, considering speed distribution directly and acceleration distribution indirectly as part of the FE speed model. In this regard, we followed the well-established trend of planning motion paths with various curves and graphs, as presented in the literature [17]. However, we obtained trajectories as integrals of the velocity plans. Unlike other studies, the proposed
2
approach has focused on using a FE grid that simultaneously ensured high accuracy and compliance with the real capabilities of the vehicle engine-drive system. In particular, the interaction between possible speeds and the respective accelerations was considered.”
Page 11 line 364: please provide more explanation on the rational for the variables used in the objective function. It seems to me that the function can be written to include more variables, so why there the speed, acceleration and sharpness the only three measures included.
Response: In the quadratic optimization model for speed plan distribution, three measures were considered (speed, acceleration, and sharpness). The two parameters (tp and Xp) characterize the average speed Vxa and are obviously not enough because many curves integrable in the interval [t0, tp] can give the same Xp. In addition, vehicle ability to increase speed is a function of the speed and depend on vehicle characteristics (i.e. acceleration is a 3D surface as a function of vehicle speed and throttle activation level). Thus, in this study it was assumed that the speed nodes are interconnected by curves over time differentiable at least twice (i.e. providing smoothness). Then, the derivatives at the nodes which reflect the slope (acceleration) and the curvature (sharpness) were included as members of the objective function along with their weights. Since this is a FE model of the curve, no other parameters except the nodal Vx, dVx/dt, and d2Vx/dt2 can be included because only the nodes need to be distributed, consistent with vehicle technical features.In addition, the selected variables have real measurable analogs in the car's sensory system, which substantially contribute to the next optimization task (tracking) which is solved after planning.
The preceding explanation was included Item 2 in the Concluding Remarks, as follows:
“In the quadratic optimization model for speed plan distribution, three measures were considered (speed, acceleration, and sharpness). The two parameters (tp and Xp) characterize the average speed Vxa and are obviously not enough because many curves integrable in the interval [t0, tp] can give the same Xp. In addition, vehicle ability to increase speed is a function of the speed and depend on vehicle characteristics (i.e. acceleration is a 3D surface as a function of vehicle speed and throttle activation level). Thus, in this study it was assumed that the speed nodes are interconnected by curves over time differentiable at least twice (i.e. providing smoothness). Then, the derivatives at the nodes which reflect the slope (acceleration) and the curvature (sharpness) were included as members of the objective function along with their weights. Since this is a FE model of the curve, no other parameters except the nodal Vx, dVx/dt, and d2Vx/dt2 can be included because only the nodes need to be distributed, consistent with vehicle technical features.“
Is it possible to conduct a sensitivity analysis of the proposed framework to understand the impact of certain problem parameters on the overall results? It would be interesting to explore the uncertainty in the output.
Response: Thank you for your valuable comment. We agree that such a task would be valuable to add to the paper. However, given that the paper is very long we have included sensitivity analysis as a future task in the Concluding Remarks (Item 6), as follows:
“In addition, a sensitivity analysis of the proposed framework can be conducted to understand the impact of certain problem parameters on the overall results, including the effect of uncertainty.”
3 We hope that you will agree.
The paper is mathematically extensive, and I must admit that some portions of the formulation was beyond my capacity. However, I am intrigued by some of the assumptions. Looking through the document it was hard to pinpoint them all. I suggest, if the authors can include a discussion of their assumptions in a separate section to allow the readers to better understand the context.
Response: Thank you for your suggestion. All assumptions are now included in a separate section (2.1 Assumptions), as follows:
“2.2 Assumptions
The developed system involves the following assumptions: (a) The roadway is assumed to be straight with ideal surface friction and no external forces such as gusting wind. In addition, the road is assumed to have very gentle vertical curvature that would not obstruct sensor measurements.
(b) Before the overtaking maneuver starts, it is assumed that the estimation of vehicles' position has already been carried out, the forecast has been made, and the passing vehicle is ready for starting the passing maneuver. In this state, the distance D0 between the passing and opposing vehicles, and the distance d0 between the impeding and passing vehicles are estimated using long and short-range radars, respectively.
(c) The length of the impeding vehicle is estimated using machine vision technology. The sensor measurements (distance and angle) are assumed to be independent.
(d) The AV is assumed to possess the input data required for the proposed model: roadway data (e.g. lane width and speed limit) and vehicle characteristics (e.g. acceleration-speed relation).
(e) Only the passing vehicle is autonomous, and the impeding and opposing vehicles are assumed to be human-driven vehicles. The overtaking maneuver involves only one impeding vehicle.
(f) In the heuristic algorithm, it is assumed that the actual speed fluctuations detected by the radars remain within the uncertainty thresholds, for which the forecast is considered reliable. Thus, the changes of vehicle locations will occur within the boundaries determined by the specified measurement thresholds. If the threshold is exceeded, the forecast should be recalculated.
(g) Uncertainty was considered in the estimation of the speeds of the passing and impeding vehicles. Based on the uncertainty thresholds, possible violations of the thresholds were analyzed. No uncertainty propagation throughout the prediction process was considered in this study.
(h) The algorithm continuously works with the vehicle sensory system within the maneuver and estimate new overtaking parameters only when thresholds are violated, thus helping to reduce the load on the on-board computer system.” One issue I would like to inquire about is how was sensory noise accounted for in the proposed algorithm? Or was it assumed that such noise did not exist. This and similar assumptions should be better documented throughout the entire paper.
4
Response: Uncertainty was considered in the estimation of the speeds of the passing and impeding vehicles. Based on the uncertainty thresholds, we analyzed possible violation of measurements thresholds (Fig. 7), which might give us answer about the allowable noise level in future tests, by assigning accessible gap for (tp - t'p). A full treatment of uncertainty and its propagation throughout the process (from input to output) is warranted as a future research topic.
This aspect was highlighted in section 2.2 Assumptions.
Minor comments:
Please remove the last paragraph in the introduction. These roadmaps represent an outdated style of writing. The information on paper organization is hardly relevant and is mostly glossed over by readers. A reviewer mentioned this to me a few years back and I have found that eliminating these roadmaps have greatly improved the flow of my own papers. I suggest that the authors do the same.
Response: Done. The paragraph was removed. We agree, this idea never crossed our minds, but we will adopt it in future papers as well (normally, the reader should have had an idea about the contents from reading the abstract).
Page 4 line 121: “The proposed system assumes a straight horizontal road, ideal friction with the road surface, and absence of additional external forces such as wind” Please consider revising this sentence. For example, “a straight horizontal road” should be “straight sections of roads” or “tangents”. Also, “ideal friction with the road surface” should be revise to “ideal surface friction” and remove the word “additional” before external factors
Response: Done. Thank you for your diligence. P
age 4 line 126: “passing vehicle is ready starting the maneuver” do you mean “passing vehicle is already starting the maneuver” or “passing vehicle is ready to start the maneuver”
Response: Done. We meant “passing vehicle is ready for starting”.
Page 4 line 130: “time moment” should be “time” only delete moment.
Response: Done.

Reviewer 2 Report
Good paper, the mathematical background is very good, on the other side, in possible scenarios of overtaking, the situation with necessity of maneuver cancellation is not clearly discussed. This scenario is critical from the safety point of view. I would like to ask the authors to describe this situation in more detail - it will certainly increase the attractiveness of this interesting paper.
Author Response
RESPONSE TO COMMENTS REVIEWER 2
Paper Title: Optimal Speed Plan for Overtaking of Autonomous Vehicles on Two-Lane Highways
Authors: Said M. Easa and Maksym Diachuk
Ms. No.: Infrastructures -779954
We would like to thank you for your thorough and most helpful comments that have helped improved the paper. Below are the responses to each comment (in blue).
Comments and Responses
Good paper, the mathematical background is very good, on the other side, in possible scenarios of overtaking, the situation with necessity of maneuver cancellation is not clearly discussed. This scenario is critical from the safety point of view. I would like to ask the authors to describe this situation in more detail - it will certainly increase the attractiveness of this interesting paper.
Response: Thank you so much for your kind comments and encouragement.
We have added an item in the Concluding Remarks to address maneuver cancellation, as follows (Item 3):
“The general idea of using the overtaking heuristic algorithm is to guarantee maneuver completion, prior to maneuver beginning, with at least the minimum safety margin. The algorithm emanates from individual vehicle operational properties. This allows to predict maximum and minimum performances for given initial conditions. Thus, considering possible changes due to random factors, for example, the algorithm will provide a decision to pass only when maneuver safely is ensured. In addition, the algorithm can foresee the situation when a vehicle must have a power margin, if the maneuver completion would require additional acceleration to prevent possible collision. Note that cancelling the overtaking maneuver is possible. However, within the framework of the presented algorithm, cancellation could happen in occasional situations (e.g. a sudden swinging of the impeding vehicle). In this case, a switch to another algorithm is needed and this obviously should be investigated in the future.”
To clarify the safety aspects of the maneuver, we have added the following item in the Concluding Remarks (Item 4):
“Determination of the required overtaking time tp is based on safety margin and adequate engine power. These criteria allow possible adjustments between power consumption and safety time depending on the situation changes and priorities. A simple, but quite effective technique, was proposed in the heuristic algorithm for finding one rational curve from an infinite number of possible realizations. That is the reason two additional derivatives (acceleration and sharpness) were used for finding the best speed distribution. All these measures aim at ensuring strict vehicle power stability and consequently safety. In addition, by considering lateral acceleration and yaw rate (indirectly) in the steering control prognosis, motion instability and possible sideslip were prevented. Thus, maneuver safety has been adequately incorporated in the proposed framework.”

Reviewer 3 Report
The authors present a method for Autonomous Vehicles to create optimal speed plans for overtaking on two-lane highways. The paper is well-written and presents interesting results in a clear and concise manner. I have actually enjoyed reading this paper.
I have the following comments for the authors that should be addressed before publication:
1) INTRODUCTION, FIRST SENTENCE: You mention that the driver's decision "is usually subjective". This is a truism/self-evident, and you could find better wording, e.g. "requires good judgement and risk evaluation, which is not always easy".
2) INTRODUCTION, SECOND SENTENCE: In the second sentence of the introduction aims to explain the "main advantage of autonomous driving", but in my view, it fails. Please improve that: why an onboard system is better than a driver making decisions? is that true? if yes, why is not implemented in all vehicles? what are the disadvantages too? This is very important to justify your research.
3) INTRODUCTION, LAST SENTENCE: As described in the last sentence of the introduction, the proposed system “assumes a straight horizontal road, ideal friction with the road surface, and absence of additional external forces such as wind.” First of all, you should add to this list of assumptions other assumptions that are currently not mentioned regarding the speed limits and the number of vehicles involved in the maneuver (only one single passing, impeding and opposing vehicle). All of these are strong assumptions, and you should acknowledge that. In particular, the sensors measurements can be particularly affected by the slope and direction of the road, being unable to measure the position and speed of opposing vehicles in various circumstances. You should elaborate on that in the introduction and the conclusion sections.
4) INTRODUCTION, GENERAL CRITICISM: If you are aiming to make an optimal speed plan for overtaking, you should not only consider how to make a safe and smooth passing maneuver, but also what is your destination, i.e. how much time are you really saving and does it worth the risk? You should acknowledge that in your manuscript.
5) FIGURE 2 (c): The phase of overtaking corresponding to maneuver completion begins when the front of the passing vehicle reaches the front of the impeding vehicle. However, in many countries the law states that the lateral movement back to the original lane should not start until the passing vehicle has already overtaken the impeding vehicle.
6) SECTION 3.1, PAGE 7, LINE 214: For safety reasons, one can argue that the passing vehicle should aim to overtake the impeding vehicle as soon as possible, to minimize the risks associated with the overtaking maneuver. Here, instead of that, you propose a geometrical method to estimate an intermediate point that satisfies your criteria of safety margin and vehicle performance reserve for unpredicted circumstances. However, I am not convinced about this: it is not clear for me what circumstances would justify to do not make the overtaking as quick as possible. You should clarify and discuss that.
7) SECTION 3.2, LAST SENTENCE: You mention that “the minimum maneuver time should also be limited based on the ratio of the times tmin and ta, tl.” Where do you present the related formulation? I could not find it – please include it if you have not.
8) FIGURE 3, 7 AND 10: If the paper is printed in B&W the reader cannot differentiate between opposing, impeding and passing vehicles.
9) FIGURES 3-7: No untis in the axis.
10) FIGURE 9: sub-plots (a) and (b) have different font sizes than (c)-(e)
Author Response
RESPONSE TO COMMENTS REVIEWER 3
Paper Title: Optimal Speed Plan for Overtaking of Autonomous Vehicles on Two-Lane Highways
Authors: Said M. Easa and Maksym Diachuk
Ms. No.: Infrastructures -779954
We would like to thank you for your thorough and most helpful comments that have helped improved the paper. Below are the responses to each comment (in blue).
Comments and Responses
The authors present a method for Autonomous Vehicles to create optimal speed plans for overtaking on two-lane highways. The paper is well-written and presents interesting results in a clear and concise manner. I have actually enjoyed reading this paper.
Response: Thank you so much for your kind comments and encouragement.
I have the following comments for the authors that should be addressed before publication:
1) INTRODUCTION, FIRST SENTENCE: You mention that the driver's decision "is usually subjective". This is a truism/self-evident, and you could find better wording, e.g. "requires good judgement and risk evaluation, which is not always easy".
Response: We indicated in the text that this concerns the case of a decision to overtake an impeding vehicle. The term "subjective" was understood to mean that despite the assisting ADAS safety systems, the responsibility remains on the driver, who can estimate a situation disregarding the on-board systems' hints. However, we agree with the remark and revised the text as follows: “On two-lane highways, the driver's decision to overtake may be risky since a driver has to judge the operational behavior of the opposing and impeding vehicles and decide to pass in a very short time. In addition, the speed plan of the passing vehicle is based on the driver’s perception of its dynamic properties.”
2) INTRODUCTION, SECOND SENTENCE: In the second sentence of the introduction aims to explain the "main advantage of autonomous driving", but in my view, it fails. Please improve that: why an onboard system is better than a driver making decisions? is that true? if yes, why is not implemented in all vehicles? what are the disadvantages too? This is very important to justify your research.
Response: Thank you for your valuable comment. The advantages of implementing autonomous vehicles along with the main challenge are described in the first paragraph, as follows: “The concept of autonomous vehicles (AV) has been under development since 1990s [1, 2], when the first field experiment conducted on a freeway was conducted in San Diego,
2
California. Autonomous vehicles have already started to appear on the roads across the globe. Clearly, as the AV market expands, transportation professionals and researchers must address an array of challenges before AV soon becomes a reality. Several government and industry entities have begun to deploy demonstrations and field tests of the technology [3-6]. Centres for testing and validation, education, products, and standards for AV have been established and a variety of conferences are being organized to discuss the planning and modeling for AV [6]. Currently, researchers, scientists, and engineers are investing significant resources to develop supporting technologies. This vibrant state of development has motivated the proposed program. Autonomous vehicles have numerous advantages [7]. Since more than 90% of fatal vehicle collisions are attributed to human error, AV technologies can substantially reduce deaths and injuries from vehicle collisions. Autonomous vehicles can also reduce traffic congestion, reduce emission of carbon dioxide, increase highway capacity, reduce fuel consumption, improve public transportation services, and provide more efficient parking. On two-lane highways, the driver's decision to overtake may be risky since a driver has to judge the operational behavior of the opposing and impeding vehicles and decide to pass in a very short time. In addition, the speed plan of the passing vehicle is based on driver’s perception of its dynamic properties.”
3) INTRODUCTION, LAST SENTENCE: As described in the last sentence of the introduction, the proposed system “assumes a straight horizontal road, ideal friction with the road surface, and absence of additional external forces such as wind.” First of all, you should add to this list of assumptions other assumptions that are currently not mentioned regarding the speed limits and the number of vehicles involved in the maneuver (only one single passing, impeding and opposing vehicle). All of these are strong assumptions, and you should acknowledge that. In particular, the sensors measurements can be particularly affected by the slope and direction of the road, being unable to measure the position and speed of opposing vehicles in various circumstances. You should elaborate on that in the introduction and the conclusion sections. Response: Thank you for your valuable suggestions. We have now created a new section (2.2 Assumptions) in which all assumptions of the proposed system are listed, as follows:
“2.2 Assumptions
The developed system involves the following assumptions: (a) The roadway is assumed to be straight with ideal surface friction and no external forces such as gusting wind. In addition, the road is assumed to have very gentle vertical curvature that would not obstruct sensor measurements.
(b) Before the overtaking maneuver starts, it is assumed that the estimation of vehicles' position has already been carried out, the forecast has been made, and the passing vehicle is ready for starting the passing maneuver. In this state, the distance D0 between the passing and opposing vehicles, and the distance d0 between the impeding and passing vehicles are estimated using long and short-range radars, respectively.
3
(c) The length of the impeding vehicle is estimated using machine vision technology. The sensor measurements (distance and angle) are assumed to be independent.
(d) The AV is assumed to possess the input data required for the proposed model: roadway data (e.g. lane width and speed limit) and vehicle characteristics (e.g. acceleration-speed relation).
(e) Only the passing vehicle is autonomous, and the impeding and opposing vehicles are assumed to be human-driven vehicles. The overtaking maneuver involves only one impeding vehicle.
(f) In the heuristic algorithm, it is assumed that the actual speed fluctuations detected by the radars remain within the uncertainty thresholds, for which the forecast is considered reliable. Thus, the changes of vehicle locations will occur within the boundaries determined by the specified measurement thresholds. If the threshold is exceeded, the forecast should be recalculated.
(g) Uncertainty was considered in the estimation of the speeds of the passing and impeding vehicles. Based on the uncertainty thresholds, possible violations of the thresholds were analyzed. No uncertainty propagation throughout the prediction process was considered in this study.
(h) The algorithm continuously works with the vehicle sensory system within the maneuver and estimate new overtaking parameters only when thresholds are violated, thus helping to reduce the load on the on-board computer system.” In addition, the Concluding Remarks now includes an item that addresses the assumptions regarding sensor measurements (Item 5), as follows: ”In this study, the roadway is assumed to be straight with ideal surface friction and no external forces (e.g. gusting wind). In addition, the road is assumed to have very gentle vertical curvature. These assumptions are necessary since sensor measurements can be particularly affected by the slope and direction of the road, being unable to measure the position and speed of the opposing vehicles in various circumstances. Fortunately, this does not represent a limitation of the model since overtaking maneuvers on two-lane highways are not permitted on sharp vertical curves because they require very long vertical curves [20, 21].”
4) INTRODUCTION, GENERAL CRITICISM: If you are aiming to make an optimal speed plan for overtaking, you should not only consider how to make a safe and smooth passing maneuver, but also what is your destination, i.e. how much time are you really saving and does it worth the risk? You should acknowledge that in your manuscript.
Response: Done. Thank you for your suggestion. The following paragraph was added at the end of the Introduction to address your comments: “The proposed method focuses on finding a rational scheme between vehicle limiting performances. The method aims at satisfying simultaneously several competing objectives, such as sufficient overtaking time, trajectory smoothness, energy consumption, and collision avoidance. The reason for implementing this approach is that if overtaking is performed with maximum vehicle performance, there will be a risk of losing lateral stability (due to random external forces) and energy consumption will be high. However, this scenario provides the
4
minimum time and reduces the probability of a head-on collision. On the other hand, if overtaking is performed slowly, good stability and controllability will be ensured, but there may not be enough safety margin at the end of the maneuver.”
5) FIGURE 2 (c): The phase of overtaking corresponding to maneuver completion begins when the front of the passing vehicle reaches the front of the impeding vehicle. However, in many countries the law states that the lateral movement back to the original lane should not start until the passing vehicle has already overtaken the impeding vehicle.
Response: Thank you for your observation. We have somewhat stipulated this condition mathematically by Eq. (32), where the planning for completing the maneuver should begin slightly later than the position of the impeding and passing vehicles at the critical point. We have explicitly added this condition in the description of Phase c, as follows: “Note that in the maneuver completion phase, the lateral movement back to the original lane will not start until the passing vehicle has already overtaken the impeding vehicle.”
6) SECTION 3.1, PAGE 7, LINE 214: For safety reasons, one can argue that the passing vehicle should aim to overtake the impeding vehicle as soon as possible, to minimize the risks associated with the overtaking maneuver. Here, instead of that, you propose a geometrical method to estimate an intermediate point that satisfies your criteria of safety margin and vehicle performance reserve for unpredicted circumstances. However, I am not convinced about this: it is not clear for me what circumstances would justify to do not make the overtaking as quick as possible. You should clarify and discuss that.
Response: Thank you for your comment. The entire movement of the autonomous vehicle is based on two types of optimization: movement planning and tracking of references (desirable planned outputs). From this perspective, overtaking should even be planned based on some initial prerequisites. In addition, when overtaking with full use of engine energy, there is no more power reserve for unforeseen circumstances (e.g. gusty wind, random forces, or control system failure). That is, in a critical situation, it is impossible to compensate for their influence, especially if at least one of the other two vehicles suddenly increases speed. Moreover, even the maximum performance modes are undesirable from the point of vehicle lateral stability and lateral sliding. If the speed is high, a stable lane change would require a greater distance according to the criterion of limited lateral acceleration. In fact, our idea is to meet safety requirements since lateral acceleration is part of the velocity distribution and control model. Moreover, the purpose of the geometric average is only to provide an adequate solution under any initial conditions. In addition, the proposed approach leads to energy saving. Our general message is that, considering safe lane change, the distribution of speed and the necessary path for the maneuver are not uniquely and linearly tied (i.e. there must be a minimum). To clarify, we can use an analogy with braking. If one brakes softly, the distance to stop will be large. But if one brakes very sharply, the distance may also be large, but we may lose
5
stability. However, when the wheel slip ratio is in the range of 10-25%, the smallest path and sufficient stability is achieved. A few sentences were added in this section to clarify and discuss the preceding aspects, as follows (Lines 276-282): ”Note that the geometric average scheme of Figure 3 justifiable from two perspectives. First, when overtaking with full use of the engine energy, there is no more power reserve for unforeseen circumstances (e.g. gusty wind, random forces, or control system failure). That is, in a critical situation, it would not be possible to compensate for their influence, especially if one (or both) of the other two vehicles suddenly increases speed. Second, the maximum performance modes are undesirable from in terms of vehicle lateral stability and lateral sliding. If the speed is high, a stable lane change would require a greater distance according to the criterion of limited lateral acceleration.”
7) SECTION 3.2, LAST SENTENCE: You mention that “the minimum maneuver time should also be limited based on the ratio of the times tmin and ta, tl” Where do you present the related formulation? I could not find it – please include it if you have not.
Response: We apologize. The conditionally maximum time (from the point of overtaking with minimum intensity) was implied. but not the minimum time of overtaking (i.e. maximum intensity). It was really a mistake in wording which was now fixed. We did not focus on this issue, since it is not critical in terms of this algorithm, and optimization and does not lead to risk in the maneuver. We only state that this time can be limited forcibly, in contrast to Point b in Fig. (2), when the danger of completing the maneuver is quite probable.
8) FIGURE 3, 7 AND 10: If the paper is printed in B&W the reader cannot differentiate between opposing, impeding and passing vehicles.
Response: The Infrastructure is online open-access journal and we believe that figures in color will help a reader to percept these figures better.
9) FIGURES 3-7: No units in the axis.
Response: Since the specified figures are schemes (not graphs) reflecting ideas, no dimensions are really needed
10) FIGURE 9: sub-plots (a) and (b) have different font sizes than (c)-(e)
Response: The Fig. 9b is a scaled Fig. 9a with emphasis on point z. Unfortunately, the font size is proportional to the axis scale, but we have diminished this figure a bit to reduce the differences.
